# Learning Robust Kernel Ensembles with Kernel Average Pooling

## Abstract

Model ensembles have long been used in machine learning to reduce the variance in individual model predictions, making them more robust to input perturbations. Pseudo-ensemble methods like *dropout* have also been commonly used in deep learning models to improve generalization. However, the application of these techniques to improve neural networks' robustness against input perturbations remains underexplored. We introduce *Kernel Average Pool (KAP)*, a new neural network building block that applies the mean filter along the kernel dimension of the layer activation tensor. We show that ensembles of kernels with similar functionality naturally emerge in convolutional neural networks equipped with *KAP* and trained with backpropagation. Moreover, we show that when combined with activation noise, KAP models are remarkably robust against various forms of adversarial attacks. Empirical evaluations on CIFAR10, CIFAR100, TinyImagenet, and Imagenet datasets show substantial improvements in robustness against strong adversarial attacks such as AutoAttack that are on par with adversarially trained networks but are importantly obtained without training on any adversarial examples.

## 1 Introduction

Model ensembles have long been used to improve robustness in the presence of noise. Classic methods like bagging (Breiman, 1996), boosting (Freund, 1995; Freund et al., 1996), and random forests (Breiman, 2001) are established approaches for reducing the variance in estimated prediction functions that build on the idea of constructing strong predictor models by combining many weaker ones. As a result, performance of these ensemble models (especially random forests) is surprisingly robust to noise variables (i.e. features) (Hastie et al., 2009).

Model ensembling has also been applied in deep learning (Zhou et al., 2001; Agarwal et al., 2021; Liu et al., 2021; Horváth et al., 2022). However, the high computational cost of training multiple neural networks and averaging their outputs at test time often quickly becomes prohibitively expensive (also see work on averaging network weights across multiple fine-tuned versions (Wortsman et al., 2022)). To tackle these challenges, alternative approaches have been proposed to allow learning pseudo-ensembles of models by allowing individual models within the ensemble to share parameters (Bachman et al., 2014; Srivastava et al., 2014; Hinton et al., 2012; Goodfellow et al., 2013). Most notably, *dropout* (Hinton et al., 2012; Srivastava et al., 2014) was introduced to approximate the process of combining exponentially many different neural networks by "dropping out" a portion of units from layers of the neural network for each batch. It was argued that this technique prevents "co-adaptation" in the neural network and leads to learning more general features (Hinton et al., 2012).

While these techniques often improve the network generalization for i.i.d. sample sets, they are not as effective in improving the network robustness against input perturbations and in particular against *adversarial attacks* (Wang et al., 2018). Adversarial attacks (Goodfellow et al., 2014), slight but carefully constructed input perturbations that can significantly impair the network's performance, are one of the major challenges to the reliability of modern neural networks. Despite numerous works on this topic in recent years, the problem remains largely unsolved (Kannan et al., 2018; Madry et al., 2017; Zhang et al., 2019; Sarkar et al., 2021; Pang et al., 2020; Bashivan et al., 2021; Rebuffi et al., 2021; Gowal et al., 2021). Moreover, the most effective empirical defense methods

against adversarial attacks (e.g. adversarial training (Madry et al., 2017) and TRADES (Zhang et al., 2019)) are extremely computationally demanding (although see more recent work on reducing their computational cost (Wong et al., 2019; Shafahi et al., 2019)).

Our central premise in this work is that *if ensembles can be learned at the level of features (in contrast to class likelihoods), the resulting hierarchy of ensembles in the neural network could potentially lead to a much more robust classifier*. To this end, we propose a simple method for learning ensembles of kernels in deep neural networks that significantly improves the network's robustness against adversarial attacks. In contrast to prior methods such as *dropout* that focus on minimizing feature co-adaptation and improving the individual features' utility in the absence of others, our method focuses on learning *feature ensembles* that form local "committees" similar to those used in Boosting and Random Forests. To create these committees in layers of a neural network, we introduce the *Kernel Average Pool (KAP)* operation that computes the average activity in nearby kernels within each layer – similar to how Average Pooling layer computes the locally averaged activity within each spatial window but instead along the kernel dimension. We show that incorporating KAP into convolutional networks leads to learning kernel ensembles that are topographically organized across the tensor dimensions over which the kernels are arranged. When combined with activation noise, these networks demonstrate a substantial boost in robustness against adversarial attacks. In contrast to other ensemble approaches to adversarial robustness, our approach does not seek to train multiple independent neural network models and instead focuses on learning kernel ensembles within a single neural network.

Our contributions are as follows:

- we introduce the kernel average pool as a simple method for learning kernel ensembles in deep neural networks.
- we demonstrate how kernel average pooling leads to learning smoothly transitioning kernel ensembles that in turn substantially improve model robustness against input noise.
- through extensive experiments on a wide range of benchmarks, we demonstrate the effectiveness of kernel average pooling on robustness against strong adversarial attacks.

## 2    RELATED WORKS AND BACKGROUND

**Adversarial attacks:** despite their superhuman performance in many vision tasks such as visual object recognition, neural network predictions are highly unreliable in the presence of input perturbations, including natural and artificial noise. While performance robustness of predictive models to natural noise have long been studied in the literature, more modern methods have been invented in the past decade to allow discovering small model-specific noise patterns (i.e. adversarial examples) that could maximize the model's risk (Goodfellow et al., 2014).

Numerous adversarial attacks have been proposed in the literature during the past decade Carlini & Wagner (2017); Croce & Hein (2020); Moosavi-Dezfooli et al. (2016); Andriushchenko et al. (2020); Brendel et al. (2017); Gowal et al. (2019). These attacks seek to find artificially generated samples that maximize the model's risk. Formally, given a classifier function $f_\theta : \mathcal{X} \rightarrow \mathcal{Y}, \mathcal{X} \subseteq \mathbb{R}^n, \mathcal{Y} = \{1, ..., C\}$, denote by $\pi(\mathbf{x}, \epsilon)$ a perturbation function (i.e. adversarial attack) which, for a given $(x, y) \in \mathcal{X} \times \mathcal{Y}$, generates a perturbed sample $x' \in \mathcal{B}(x, \epsilon)$ within the $\epsilon$-neighborhood of $x$, $\mathcal{B}(x, \epsilon) = \{\mathbf{x}' \in \mathcal{X} : \|x' - x\|_p < \epsilon\}$, by solving the following maximization problem

$$\max_{t \in \mathcal{B}(x,\epsilon)} \mathcal{L}(f_\theta(t), y), \tag{1}$$

where $\mathcal{L}$ is the classification loss function (i.e. classifier's risk) and $\|.\|_p$ is the $L_p$ norm function. Solutions $\mathbf{x}'$ are called *adversarial examples* and are essentially the original input samples altered with additive noise of magnitude $\epsilon$ measured by the $L_p$ norm.

**Adversarial defenses:** Concurrent to the research on adversarial attacks, numerous methods have also been proposed to defend neural network models against these attacks (Kannan et al., 2018; Madry et al., 2017; Zhang et al., 2019; Sarkar et al., 2021; Pang et al., 2020; Bashivan et al., 2021; Robey et al., 2021; Sehwag et al., 2022; Rebuffi et al., 2021; Gowal et al., 2021). Formally, the goal of these defense methods is to guarantee that the model predictions match the true label not only over the sample set but also within the $\epsilon$-neighborhood of samples $\mathbf{x}$. Adversarial training,

which is the most established defense method to date, formulates adversarial defense as a minimax optimization problem through which the classifier's risk for adversarially perturbed samples is iteratively minimized during training (Madry et al., 2017). Likewise, other prominent methods such as ALP (Kannan et al., 2018) and TRADES (Zhang et al., 2019), encourage the classifier to predict matching labels for the original ($\mathbf{x}$) and perturbed samples ($\mathbf{x}'$).

Despite the continuing progress towards robust neural networks, most adversarial defenses remain computationally demanding, requiring an order of magnitude or more computational resources compared to normal training of these networks. This issue has highlighted the dire need for computationally cheaper defense methods that are also scalable to large-scale datasets such as Imagenet. In that regard, several recent papers have proposed alternative methods for discovering diverse adversarial examples at a much lower computational cost and have been shown to perform competitively with adversarial training using costly iterative attacks like Projected Gradient Descent (PGD) (Wong et al., 2019; Shafahi et al., 2019).

Another line of work has proposed utilizing random additive noise as a way to empirically improve the neural network robustness (Liu et al., 2018; Wang et al., 2018; He et al., 2019) and to derive robustness guarantees (Cohen et al., 2019; Lecuyer et al., 2019). Although, some of the proposed defenses in this category have later been discovered to remain vulnerable to other forms of attacks (Tramer et al., 2020), there is an increasing body of work that shows the close relationship between robustness to random noise and adversarial robustness (Ford et al., 2019; Cohen et al., 2019). Also related to our proposed method, recent work on feature denoising (Xie et al., 2019) shows that denoising feature maps in neural networks together with adversarial training leads to large gains in robustness against adversarial examples. However, this work is fundamentally different from our proposed method in that the focus of this work is on *denoising* individual feature maps by considering the distribution of feature values across the spatial dimensions within each feature map.

**Ensemble methods:** Ensemble methods have long been used in machine learning and deep learning because of their effectiveness in improving generalization and obtaining robust performance against input noise (Hastie et al., 2009). In neural networks, pseudo-ensemble methods like *dropout* Hinton et al. (2012) create and simultaneously train an ensemble of "child" models spawned from a "parent" model using parameter perturbations sampled from a perturbation distribution (Bachman et al., 2014). Through this procedure, pseudo-ensemble methods can improve generalization and robustness against input noise. Another related method is MaxOut Goodfellow et al. (2013) which proposes an activation function that selects the maximum output amongst a series of unit outputs.

Naturally, similar ideas consisting of neural network ensembles have been tested in recent years to improve prediction variability and robustness in neural networks with various degrees of success (Pang et al., 2019; Kariyappa & Qureshi, 2019; Abbasi et al., 2020; Horváth et al., 2022; Liu et al., 2021). Ensemble adversarial training (Tramèr et al., 2018) proposes to use adversarial examples transferred from various models during adversarial training to improve the robustness of the model. Several other works have focused on enhancing the diversity among models within the ensemble with the goal of making it more difficult for adversarial examples to transfer between models (Pang et al., 2019; Kariyappa & Qureshi, 2019). However these ensemble models still remain prone to ensembles of adversarial attacks (Tramer et al., 2020).

## 3 METHODS

### 3.1 PRELIMINARIES

Let $f_\theta(\mathbf{x}) : \mathcal{X} \rightarrow \mathcal{Y}$, where $\mathcal{X} \subseteq \mathbb{R}^n$, $\mathcal{Y} = \{1, ..., C\}$, be a classifier with parameters $\theta$. In feed-forward deep neural networks, the classifier $f_\theta$ is usually composed of a cascade of simpler functions $f^{(l)}(\mathbf{x})$, $l \in \{1, \ldots, L\}$ chained together such that the network output is computed as $y = f^{(L)}(f^{(L-1)}(\ldots f^{(1)}(\mathbf{x})))$. For our function $f_\theta$ to correctly classify the input patterns $\mathbf{x}$, we wish for it to attain a small risk for $(\mathbf{x}, y) \sim \mathcal{D}$ as measured by loss function $\mathcal{L}$. Additionally, for our classifier to be robust, we also wish $f_\theta$ to attain a small risk in the vicinity of all $\mathbf{x} \in \mathcal{X}$, normally defined by a Wasserstein ball around the sample points (Madry et al., 2017).

While to guarantee robustness, one has to consider the maximum risk within the epsilon ball, in practice, the prediction variance can arguably be also linked to the expected robustness – similar to recent work on domain generalization that uses risk variance as an objective for improving model

generalization across domains (Krueger et al., 2021). Intuitively, a model which has a high prediction variance (or similarly high risk variance) to noisy inputs, is more likely to exhibit extreme high risks for data points sampled from the same distribution (i.e. adversarial examples). Indeed, classifiers that generate lower variance predictions are often expected to generalize better and be more robust to input noise. For example, classic ensemble methods like *bagging*, *boosting*, and *random forests* operate by combining the decisions of many weak (i.e. high variance) classifiers into a stronger one with reduced prediction variance and improved generalization performance (Hastie et al., 2009).

Given an ensemble of predictor functions $f_i, i \in 1, \ldots, K$ with zero or small biases, the ensemble prediction (normally considered as the mean prediction $\bar{y} = \frac{1}{K} \sum_{i=1}^{K} y_i$) reduces the expected generalization loss by shrinking the prediction variance. To demonstrate the point, one can consider $K$ i.i.d. random variables with variance $\sigma^2$ and their average value that has a variance of $\frac{\sigma^2}{K}$. Based on this logic, one would expect ensembles of neural network classifiers to be more robust in the presence of noise or input perturbations in general. However, such ensemble models have been shown to remain prone to ensemble of adversarial attacks with large epsilons (Tramer et al., 2020).

We reasoned that individual networks participating in these ensembles may still learn different sets of non-robust representations leaving room for the attackers to find common weak spots across all individual models within the ensemble. At the same time, constructing ever-larger ensemble classifiers might quickly become infeasible, especially in the case of neural network classifiers. On the other hand, learning robust features has been suggested as a way towards robust classification (Bashivan et al., 2021). Consequently, if individual kernels within a single network are robust, it would become much more difficult to find adversaries that can fool the full network. In the next section, we introduce the *Kernel Average Pool* as a way towards learning ensemble kernels with better robustness properties against input perturbations.

## 3.2 KERNEL AVERAGE POOL (KAP)

Mean filters (a.k.a., average pool) are widely accepted as simple noise suppression mechanisms in computer vision. Spatial average pooling layers are commonly used in modern deep neural networks (Zoph et al., 2018) by applying a mean filter along the spatial dimensions of the input to reduce the effect of spatially distributed noise (e.g. adjacent pixels in an image).

Here, we wish to substitute each kernel in the neural network model with an ensemble of kernels performing the same function such that the ensemble output is the average of individual kernel outputs. This can be conveniently carried out by applying the average pool operation along the kernel dimension of the input tensor.

Given an input $z \in \mathbb{R}^{D \times N_k}$, where $D$ and $N_k$ denote the input dimension and the number of kernels respectively, the kernel average pool operation with kernel size $K$ and stride $S$, computes the function

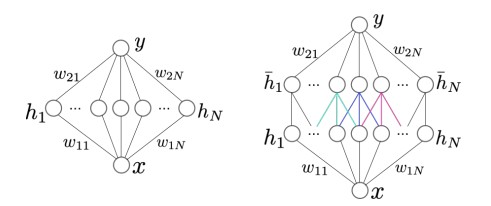

Figure 1: Schematic of a one-layer neural network with (right) and without (left) the kernel average pooling operation.

$$\bar{z}_{ik} = \frac{1}{K} \sum_{l=Sk-\frac{K-1}{2}}^{Sk+\frac{K-1}{2}} z_{il} \tag{2}$$

Importantly, when $z$ is the output of an operation linear with respect to the weights on an input $x$ (e.g. linear layers or convolutional layers), KAP is functionally equal to computing the locally averaged weights within the layer and could be interpreted as a form of Kernel Smoothing (Wang et al., 2020) conditioned that the nearby kernels are more or less similar to each other.

$$\bar{z}_{ik} = \frac{1}{K} \sum_{l=Sk-\frac{K-1}{2}}^{Sk+\frac{K-1}{2}} w_i x = \left( \frac{1}{K} \sum_{l=Sk-\frac{K-1}{2}}^{Sk+\frac{K-1}{2}} w_i \right) x \tag{3}$$

Moreover, the degree of overlap (i.e. parameter sharing) across kernel ensembles can be flexibly controlled by adjusting the KAP stride. Choosing stride $S = K$, produces independent kernel en-

sembles (no parameter sharing between ensembles), while having stride $S < K$ enforces parameter sharing across kernel ensembles.

Eq. 2 assumes that kernels are arranged along one tensor dimension. However, KAP could more generally be applied on any D-dimensional tensor arrangement of kernels. For example to apply a 2-dimensional KAP (mainly considered in our experiments) on the input $\mathbf{z}$, one can first reshape the channel dimension into a 2-dimensional array and then apply KAP along the two kernel dimensions (see §A.1 and Alg. 1). Importantly, using higher dimensional tensor arrangements of kernels when applying KAP, allows parameter sharing across a larger number of kernel ensembles.

---

**Algorithm 1** 2D Kernel Average Pool

**Input:** layer input $\mathbf{x}$, kernel size $k$, stride $s$, vectorization operator *Vec* and its inverse $Vec^{-1}$, zero padding function *Pad*, and average pooling function *AvePool*.

$\mathbf{h} \leftarrow Vec^{-1}_{\sqrt{D}, \sqrt{D}, WH}(Vec(\mathbf{x}^T))$

$\mathbf{h} \leftarrow Pad(\mathbf{h}, \frac{k-1}{2})$

$\mathbf{h} \leftarrow AvePool(\mathbf{h}, k, s)$

$\mathbf{h} \leftarrow Vec^{-1}_{W,H,D}(Vec(\mathbf{h}^T))$

**return:** $\mathbf{h}$

---

In the next two subsections, we will explain how training networks with KAP-layers leads to learning topographically organized kernel ensembles (§3.3) and how these kernel ensembles may contribute to model robustness (§3.4).

## 3.3 KERNEL AVERAGE POOLING YIELDS TOPOGRAPHICALLY ORGANIZED KERNEL ENSEMBLES

Consider a simple neural network with one hidden layer and $N_k$ units (Fig.1-left) where the hidden unit activation is $h_i = w_{1i}x$ and the network output $y$ is computed as

$$y = \sum_{i=1}^{N_k} w_{2i}h_i \tag{4}$$

In this network, the output gradients with respect to weight parameters can be computed as

$$\frac{\partial y}{\partial w_{1i}} = w_{2i}x, \qquad \frac{\partial y}{\partial w_{2i}} = w_{1i}x \tag{5}$$

Now consider a variation of this network where the hidden unit activations are passed through a kernel average pool with kernel size $K$ (Fig.1-right). In the KAP-network, the output gradients with respect to weight parameters are altered such that

$$\frac{\partial y}{\partial w_{1i}} = \frac{1}{K} \sum_{l=i-\frac{K-1}{2}}^{i+\frac{K-1}{2}} w_{2l}x, \qquad \frac{\partial y}{\partial w_{2i}} = \frac{1}{K} \sum_{l=i-\frac{K-1}{2}}^{i+\frac{K-1}{2}} w_{1l}x \tag{6}$$

where, to simplify the limits, $K$ is assumed to be an odd number. In contrast to the regular network, in the KAP-network, the gradients of the output with respect to the incoming ($w_{1i}$) and outgoing ($w_{2i}$) weights in node $i$ depend on the average of outgoing and incoming weights, respectively, over the kernel average pool window. The difference in the output gradients with respect to weights $w_{1i}$ (and similarly for $w_{2i}$) for nodes $i$ and $j = i + d$ can be written as

$$\frac{\partial y}{\partial w_{1i}} - \frac{\partial y}{\partial w_{1j}} = \sum_{l=i-\frac{K-1}{2}}^{i+\frac{K-1}{2}} w_{2l}x - \sum_{l=i+d-\frac{K-1}{2}}^{i+d+\frac{K-1}{2}} w_{2l}x \tag{7}$$

From Eq.7, it is clear that when $K$ is large $K \gg 1$, the difference in the output gradients with respect to weights for a pair of nodes $(i, j)$ depends on the absolute difference between node indices ($|i - j|$) and is smaller for a pair of nodes with smaller index difference (i.e. physically closer nodes). Thus, when training with backpropagation, weights connected to physically closer nodes (in

terms of their index numbers) will likely receive more similar gradients compared to those that are farther. Consequently, these physically closer weights are more likely to converge to more similar values. On the other hand, for a KAP with stride=$S$ and kernel size $K$, each kernel participates in $\lceil \frac{K^2}{S} \rceil^2$ ensembles. Kernel sharing between ensembles provides a natural mechanism preventing the participating kernels in each group from converging to the exact same parameter values. Our empirical results show that the interaction between these two forces leads to smoothly transitioning topographically organized kernel maps (Fig.3).

### 3.4 KERNEL AVERAGE POOLS AND ADVERSARIAL ROBUSTNESS

Our approach is closely related to randomized smoothing (Lecuyer et al., 2019; Cohen et al., 2019), which for a sample $(x, y)$ consists of learning to predict $y$ with higher probability than other classes over $\mathcal{N}(x, \sigma^2 I)$. When combined with noise resampling to estimate the true class, randomized smoothing yields certifiable robustness against adversarial perturbations. More recently, Horváth et al. (2022) show that in addition to injecting noise in the input, averaging the logits over an ensemble of models leads to a reduction of the variance due to the noisy inputs in randomized smoothing and in turn improves the certified robustness radius. They propose to learn an ensemble

$$g(x) = Ens(f_c(x + n)) = \frac{1}{L} \sum_l f_c^{(l)}(x + n) \tag{8}$$

where $n \sim \mathcal{N}(0, \sigma^2 I)$ and $f_c^{(l)}$ denotes the $l$-th presoftmax classifier in an ensemble of $L$ classifiers.

Similar to randomized smoothing (Lecuyer et al., 2019; Cohen et al., 2019), by introducing stochasticity in the input during training as in randomized smoothing, we hope to learn a model that is robust to perturbations. Moreover, in a simple setting where only one layer of KAP is used without activations, we note that KAP averages multiple filters before applying them to the input of a KAP block (eq. 3). This can be understood as averaging the features obtained from multiple models, each corresponding to a filter. Unlike Horváth et al. (2022), our ensembling is done at the level of features instead of the logits. Nevertheless, their arguments still apply to the case of a reduction of variance in the features computed. Therefore, if KAP features are used as inputs to fully connected layers to compute logits, a reduction of variance in the features will translate into a reduction of variance in the logits. An additional key difference is that in our case, in the full KAP-based models used in §4, we use networks consisting of multiple cascaded KAP blocks to extract features, similar to how multiple convolutional blocks are used sequentially in convolutional networks. Each KAP block consists of an ensembling, from an input that was perturbed with Gaussian noise. In other words, our approach can be understood as recursively performing a form of Horváth et al. (2022)'s randomized smoothing with ensembles, by interpreting the output of each KAP block as a randomized smoothing input for the next block. In this light, our KAP architectures perform the operation

$$f_c \circ g_{N_l} \circ ... \circ g_1(x) \text{ where } g_i(x) = Ens[f_i(x + n_i)] \tag{9}$$

where the ensembling is performed by the KAP operation, potentially including activation functions, $N_l$ is the number of KAP blocks, $n_i$ is Gaussian noise sampled in KAP block $i$, $f_i$ is the operation (e.g. a convolution) performed in KAP block $i$ before a kernel average pool, and $f_c$ maps the representations to the logits of the classes. Our reasoning for this recursive approach is twofold: first, during training, this encourages all layers to learn to be robust to variance in their input. Otherwise in deep networks, some layers may not be exposed to significant variance in their input depending on their depth, due to the variance reduction occurring in the earlier layers. Second, we hope that by having randomized smoothing at every KAP block, the model will require less perturbed inputs. If this approach successfully performs randomized smoothing on the features, we may hope that this will lead the representations of adversarially perturbed inputs to remain close to the distribution of inputs perturbed with Gaussian noise, on which the network has been trained and therefore should perform well. We empirically verify this intuition in the appendix (Fig.A4,A5).

## 4 EXPERIMENTS

In this section we empirically demonstrate the effectiveness of our proposed kernel average pool operation in boosting robustness in deep neural networks.

## 4.1 EXPERIMENTAL SETUP

**Datasets:** We validated our proposed method on several standard benchmarks CIFAR10, CIFAR100 (Krizhevsky, 2009), TinyImagenet Le & Yang (2015), and Imagenet (Deng et al., 2009). We used standard preprocessing of images consisting of random crop and horizontal flipping for all datasets. We used images of size 32 in our experiments on CIFAR datasets, 128 for TinyImagenet, and 224 for Imagenet.

**Baseline models:** We compared the results from our proposed model to several baselines including (i) the original ResNet18 (i.e. without additional KAPs) trained normally (marked with **NT**); (ii) the original ResNet18 with additive activation noise (marked with **NT**($\sigma$ = .)); (iii) the original ResNet18 trained using adversarial training with early stopping (marked with **AT-ES**).

**Adversarial attacks:** We assessed the model robustness against various adversarial attacks including $L_\infty$ Projected Gradient Descent (PGD) (Madry et al., 2017), SQUARE black-box (Andriushchenko et al., 2020) attack and the AutoAttack (Croce & Hein, 2020). Notably, AutoAttack is an ensemble attack consisting of four independent attacks and is considered as one of the strongest adversarial attacks in the field. We used $\epsilon$ value of 0.031 for experiments on CIFAR datasets and 0.016 for TinyImagenet and Imagenet datasets. See supplementary Table A3 for full details of attacks used for model evaluation. Importantly, we applied each attack on the model without activation noise to prevent activation noise from potentially masking the gradients in the network.

**Training and evaluation considerations:** In our experiments, we primarily used the ResNet18 architecture (He et al., 2016) that consists of four groups of layers and each group containing two basic residual blocks. For the KAP variations of ResNet18, we added the KAP operation after each convolution in the network. In variations of the model where we introduced activation noise, we added random noise sampled from the Gaussian distribution $\mathcal{N}(0, \sigma^2)$ after each KAP operation (and after each convolution in the original model architecture).

For adversarial training of the baseline **AT** models on CIFAR10, CIFAR100, and TinyImagenet datasets, we used the normal adversarial training procedure with early stopping and $L_\infty$ PGD attack (Madry et al., 2017; Rice et al., 2020). We used 20 iterations for CIFAR training runs and 10 iterations for TinyImagenet. On Imagenet dataset, we used Fast-AT method (Wong et al., 2019) with the default training parameters consisting of three training phases with increasing image resolution.

## 4.2 ROBUSTNESS AGAINST STANDARD ADVERSARIAL ATTACKS ON CIFAR

Table 1: Comparison of adversarial accuracy against various attacks on CIFAR10 and CIFAR100 datasets. For all attacks we used $\epsilon = 0.031$. All attacks are performed on the corresponding model without input or activation noise.

| DATASET | MODEL | CLEAN | PGD-$L_\infty$ | AUTOATTACK | SQUARE |
|---------|-------|-------|-------|-------|-------|
| CIFAR10 | RN18-NT | 94.66 | 0.0 | 0.0 | 0.87 |
| | RN18-NT ($\sigma = 0.1$) | 88.95 | 9.69 | 8.90 | **61.7** |
| | WRN-16-4(BE) (WEN ET AL., 2020) | 95.60 | 7.90 | 7.80 | 21.00 |
| | RN18-AT-ES (RICE ET AL., 2020) | 84.20 | 43.70 | 43.00 | 49.10 |
| | RN18-KAP-NT ($\sigma = 0.1, K = 3$) | 79.09 | **67.7** | 41.8 | 44.49 |
| | RN18-KAP-NT ($\sigma = 0.2, K = 3$) | 74.30 | 65.1 | **44.40** | 47.5 |
| CIFAR100 | RN18-NT | 74.00 | 0.0 | 0.0 | 0.20 |
| | RN18-NT ($\sigma = 0.1$) | 61.60 | 6.00 | 5.20 | **33.70** |
| | RN18-AT-ES (RICE ET AL., 2020) | 56.50 | 20.40 | **19.60** | 22.86 |
| | RN18-KAP-NT ($\sigma = 0.1, K = 3$) | 48.20 | **33.9** | 15.70 | 17.40 |
| | RN18-KAP-NT ($\sigma = 0.2, K = 3$) | 38.60 | 31.9 | 17.60 | 27.40 |

We first compared robustness in convolutional neural networks with and without KAP on the CIFAR10 and CIFAR100 datasets. For this we trained the vanilla ResNet18 architecture and two variations of this architecture where all convolution operations were followed by KAP. To make sure that the stochasticity due to activation noise does not interfere with gradient estimation during attacks, we performed all attacks on the corresponding model without input or activation noise.

Table 1 lists the robustness of different models trained on CIFAR10 and CIFAR100 datasets against several commonly used adversarial attacks. Confirming prior work (He et al., 2019), we found that on both CIFAR10 and CIFAR100 datasets, training the network with noisy activations can improve

the network robustness. This improvement was most noticeable against the strong SQUARE black-box attack and to a lesser degree against PGD and AutoAttack. Furthermore, we found that adding KAP to the model significantly improves the robustness against all attacks and even beyond that of the adversarially trained ResNet18 architecture (**AT**). Robustness remained high against adversarial examples that were generated using models with activation noise (§A.2) as well as those transferred from RN18-NT and RN18-AT models (TableA5). The improved robustness was however at the expense of a noticeable reduction in clean accuracy that also increased with the activation noise variance $\sigma$. Although, it should also be noted that KAP models were trained using normal training procedures (i.e., no adversarial training) and as a result the computational cost of training was only a fraction of that required for adversarial training ($\sim 0.14\%$). See the supplementary Table A4 for a comparison of training speed between KAP models and adversarial training.

We additionally explored the effect of kernel pooling type (No pooling, Max pooling, and Average pooling) and activation noise variance ($\sigma \in \{0, 0.1, 0.2\}$) on the robust accuracy with the ResNet18 architecture. To get a better sense of how robustness generalizes to higher-strength attacks (i.e. larger $\epsilon$), we also tested each model on several $\epsilon$ values ($\frac{2}{255}$ to $\frac{32}{255}$) using AutoAttack (Fig. 2). We found that (a) adding KAP to RN18 without activation noise already makes the network substantially more robust against attacks with smaller epsilons (Fig. 2a); (b) KAP model showed strong robustness to adversarial attacks on par with RN18-AT and even better performance against attacks with larger $\epsilon$ (Fig. 2b), while the variation with Kernel Max Pool was the least robust variation; c) larger activation noise variance during training led to higher robustness against stronger attacks (Fig. 2c). Furthermore, in separate experiments, we also investigated the effect of model depth and kernel ensemble size on robustness, which we report in the appendix §A.3. We found that increasing the network depth and KAP kernel size both substantially improve the network robustness to AutoAttack.

### 4.3 Robustness against adversarial attacks on Imagenet

To test whether our results scale to larger datasets, we also trained and compared convolutional neural networks on two large-scale datasets, namely TinyImagenet (200 classes) and Imagenet (1000 classes). Here again, we used the ResNet18 architecture as our baseline and created variations of this architecture by adding KAP after every convolution operation. We found that reducing the weight decay parameter when training the KAP networks on Imagenet improves the performance of the KAP models and used a weight decay of $1e^{-5}$ in training our best models on Imagenet. In addition to the PGD-$L_\infty$ attack, we also evaluated the models using AutoAttack on these datasets. However, because of its high computational cost, we used 1000 random samples from the validation set.

Table 2: Comparison of robust accuracy against various attacks on TinyImagenet and Imagenet datasets. For all attacks we used $\epsilon = 0.016$. All attacks are performed on the corresponding model without input or activation noise. †: models trained using Fast Adversarial Training (Wong et al., 2019).

| Dataset | Model | Clean | PGD-$L_\infty$ | AutoAttack | Square |
|---|---|---|---|---|---|
| | RN18-NT | 58.90 | 0.00 | 0.00 | 3.30 |
| TinyImagenet | RN18-NT ($\sigma = 0.1$) | 56.50 | 1.90 | 1.80 | **35.50** |
| | RN18-AT-ES (Rice et al., 2020) | 45.80 | **25.40** | **21.60** | 29.10 |
| | RN18-KAP-NT ($\sigma = 0.1, K = 3$) | 39.60 | **25.70** | 16.5 | 18.70 |
| | RN18-NT | 68.68 | 0.0 | 0.0 | 2.80 |
| | RN18-NT ($\sigma = 0.1$) | 51.20 | 0.19 | 0.20 | 6.70 |
| Imagenet | RN18-AT† | 53.20 | 8.00 | 8.00 | 8.20 |
| | RN18-KAP-NT ($\sigma = 0.1, K = 3$) | 9.60 | 6.05 | 2.85 | 2.95 |
| | RN18WideX4-AT† | 62.00 | 27.81 | 11.80 | **14.10** |
| | RN18WideX4-KAP-NT ($\sigma = 0.1, K = 3$) | 38.00 | **31.49** | **15.3** | **14.40** |

Table 2 summarizes the robust accuracy of different models on these two datasets. Overall, we found that (a) on these two datasets, sole usage of the activation noise similar to (He et al., 2019) was much less effective at improving the network robustness; (b) robustness in the KAP variation of ResNet18 was significantly better than the orignal network and baseline trained with noise but slightly lower than the adversarial trained network on TinyImagenet dataset. On Imagenet, we found that the KAP variation of ResNet18 model was struggling to learn the task completely, reaching only about 10-12% accuracy on the clean dataset. We reasoned that the difficulty in learning the task on this dataset might be due to the large number of classes in this dataset and the possibility that the ResNet18

model does not have sufficient capacity to learn enough kernel ensembles to tackle this dataset. For this reason we also trained a wider version of RN18 in which we multiplied the number of kernels in each layer by a factor of 4 (dubbed **RN18WideX4**). We found that this wider network significantly boosted the robust accuracy on Imagenet, even surpassing the adversarially trained model. However, again we found that this boost to adversarial robustness is accompanied by a significant decrease in clean accuracy.

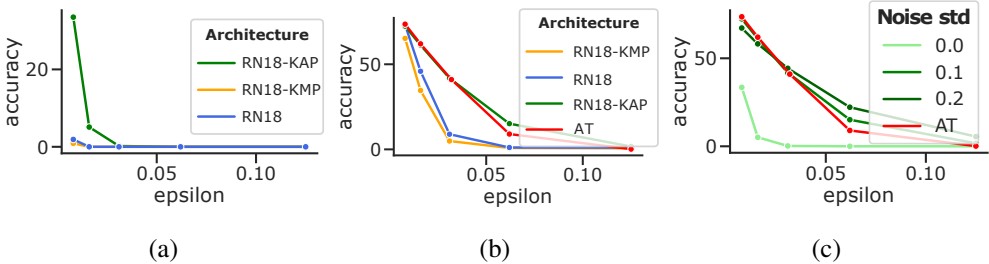

(a)          (b)          (c)

Figure 2: Robust accuracy in RN18-KAP model against AutoAttack with various attack strength $\epsilon$ on CIFAR10 dataset. (a) RN18 and RN18-KAP ($\sigma = 0$, $K = 3$); (b) RN18, RN18-KAP, RN18-KMP ($\sigma = 0.1$, $K = 3$) and AT; (c) RN18-KAP ($K = 3$) for various noise levels $\sigma$.

### 4.4 KAP YIELDS TOPOGRAPHICALLY ORGANIZED KERNELS

As described in the methods, KAP combines the output of multiple kernels into a single activation passed from one layer to the next. As this operation creates dependencies between multiple kernels within each group, we expected a certain level of similarity between the kernels within each pseudo-ensemble to emerge. To examine this, we visualized the weights in the first convolutional layer of the normally trained (**NT**), the adversarially trained RN18 architecture (**AT**) and two variations of RN18 with Kernel Average Pool (**KAP**) and Kernel Max Pool (**KMP**) on CIFAR10 and Imagenet datasets (Fig. 3). As a reminder for these experiments, we had incorporated a 2-dimensional KAP that was applied on the kernels arranged on a 2-dimensional sheet.

As expected, the kernels in **NT** and **AT** models did not show any topographical organization. Kernels in **KMP** model were sparsely distributed, with many kernels containing very small weights. This was presumably because of the competition amongst different kernels within each pseudo-ensemble that had driven the network to ignore many of its kernels. In contrast, in **KAP** model, we observed an overall topographical organization in the arrangement of the learned kernels along the two dimensional sheet. Moreover, in many cases the kernels gradually shifted from the dominant pattern in one cluster to another as traversing along either of the two kernel dimensions. This topographical organization of the kernels on the 2-dimensional sheet is reminiscent of the topographical organization of the orientation selectivity of neurons in the primary visual cortex of primates (Hubel & Wiesel, 1977).

| C10-NT | C10-KMP | C10-AT | C10-KAP | Imagenet-AT | Imagenet-KAP |
|--------|---------|--------|---------|-------------|--------------|

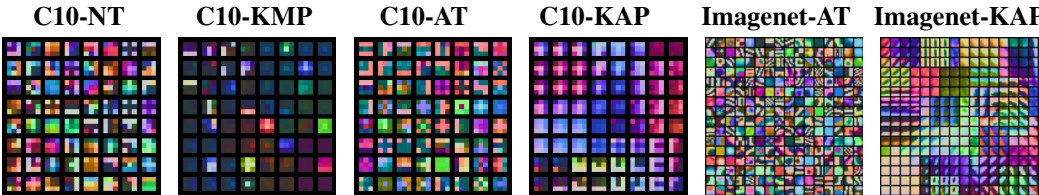

Figure 3: Visualization of the learned weights in the first layer of several variations of RN18 model.

## 5 CONCLUSION

We proposed Kernel Average Pooling as a mechanism for learning ensemble of kernels in layers of deep neural networks. We showed that when combined with activation noise, KAP-networks form a process that can be thought of as recursive randomized smoothing with ensembles applied at the level of features, where each stage consists of applying ensemble of kernels followed by noise injection. Our empirical results demonstrated significant improvement in network robustness

at a fraction of computational cost of state-of-the-art methods like adversarial training. However, because of the need for learning ensemble of kernels at each network layer, the improved robustness is often accompanied by reduced performance on the clean datasets. Our results suggest feature-level ensembling as a practical and scalable approach for training robust neural networks.

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

# A    APPENDIX

## A.1    2-DIMENSIONSAL KAP

Given an input $z \in \mathbb{R}^{D \times N_k}$, where $N_k = N_r \times N_c$ is the number of kernels that can be rearranged into $N_r$ rows and $N_c$ columns, and $D$ denotes the input dimension, the 2D kernel average pool operation with kernel size $K \times K$ and stride $S$, computes the function

$$\bar{z}_{ik}(x) = \frac{1}{K^2} \sum_{l=\lfloor \frac{Sk}{N_c} \rfloor - \frac{K-1}{2}}^{\lfloor \frac{Sk}{N_c} \rfloor + \frac{K-1}{2}} \sum_{m=(Sk \bmod N_c) - \frac{K-1}{2}}^{(Sk \bmod N_c) + \frac{K-1}{2}} z_{i(lN_c+m)} \tag{10}$$

Graphically, this procedure is visualized in Fig. A1.

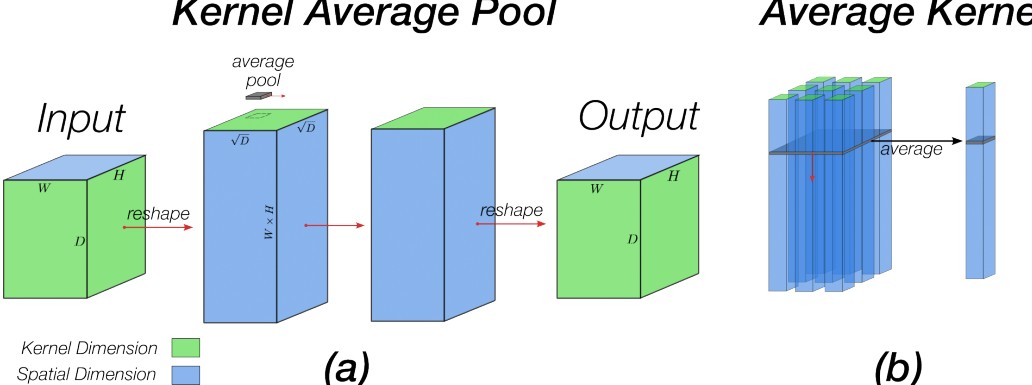

Figure A1: Graphic illustration of 2-dimensional KAP. a) the input tensor is first reshaped such that the spatial dimensions are collapsed onto a single dimension and the kernel dimension is rearranged as a matrix. A 2D average pool is applied on the reshaped tensor, and the resulting tensor is reshaped back into its original shape; b) the average kernel is applied per spatial position (i.e. a pixel) and computes the average of nearby kernel values at that spatial position.

## A.2    CLASSIFICATION ROBUSTNESS FOR DIRECT ATTACKS ON FULL MODELS WITH ACTIVATION NOISE

For completeness of our assessments, we also considered the case where the attacker is applied to the full model with activation noise. Tables A1 and A2 summarize the robust accuracies on CIFAR10, CIFAR100, TinyImagenet, and Imagenet datasets. Compared to the case where the attackers were applied on the model without activation noise, the KAP models achieved higher robust accuracies against attacks performed on the full model.

We reasoned that this improvement in robustness is potentially because of the test-time stochasticity in layer activations and its possible detrimental effect on gradient estimation in attacks that only use a single sample. To further investigate this, we also tested all of our models against the *random* version of AutoAttack ($AutoAttack_{rnd}$) in addition to the standard version of AutoAttack ($AutoAttack_{std}$). This attack applies Expectation over Transformation introduced by (Athalye et al., 2018) to correctly compute the gradients over the expected transformation to the input (apgd-ce and apgd-dlr attacks, each with 20 iterations). We found that KAP models were also equally robust against the *random* version of AutoAttack.

## A.3    EFFECT OF NETWORK DEPTH AND KERNEL ENSEMBLE SIZE ON NETWORK ROBUSTNESS

We performed three additional experiments to investigate the effect of network depth, number of KAP layers, and kernel ensemble size (which is controlled by KAP kernel size) on the network robustness. In the first experiment, we varied the network depth (number of convolutional blocks)

Table A1: Comparison of adversarial accuracy against various attacks on CIFAR10 and CIFAR100 datasets where each attack is performed on the full model with activation noise. For all attacks we used $\epsilon = 0.031$. AutoAttack$_{rnd}$ results for stochastic models are reported for 20 and 50 (in parentheses) EoTs.

| Dataset | Model | Clean | PGD-$L_\infty$ | AutoAttack$_{std}$ | AutoAttack$_{rnd}$ | Square |
|---|---|---|---|---|---|---|
| | RN18-NT | 94.66 | 0.0 | 0.0 | 0.0 | 0.87 |
| | RN18-NT ($\sigma = 0.1$) | 88.95 | 10.91 | 8.79 | 8.30 (6.20) | **84.42** |
| CIFAR10 | RN18-AT-ES | 84.20 | 43.70 | 43.00 | 43.38 | 49.10 |
| | RN18-KAP-NT ($\sigma = 0.1, K = 3$) | 79.09 | **67.55** | **58.40** | 61.64 (56.1) | 66.12 |
| | RN18-KAP-NT ($\sigma = 0.2, K \in \{3\}$) | 74.30 | 61.82 | 55.19 | **57.04 (57.80)** | 63.64 |
| | RN18-NT | 74.00 | 0.0 | 0.0 | 0.0 | 0.24 |
| | RN18-NT ($\sigma = 0.1$) | 61.60 | 5.23 | 4.06 | 4.30 (4.40) | **49.46** |
| CIFAR100 | RN18-AT-ES | 56.50 | 20.40 | 19.60 | 19.88 | 22.86 |
| | RN18-KAP-NT ($\sigma = 0.1, K \in \{3\}$) | 48.20 | **36.08** | **23.70** | 30.52 (28.20) | 29.80 |
| | RN18-KAP-NT ($\sigma = 0.2, K \in \{3\}$) | 38.60 | 31.49 | 23.36 | **31.20 (31.00)** | 27.40 |

Table A2: Comparison of robust accuracy against various attacks on TinyImagenet and Imagenet datasets where each attack is performed on the full model with activation noise. For all attacks we used $\epsilon = 0.016$. †: models trained using Fast Adversarial Training (Wong et al., 2019).

| Dataset | Model | Clean | PGD-$L_\infty$ | AutoAttack$_{std}$ | AutoAttack$_{rnd}$ |
|---|---|---|---|---|---|
| | RN18-NT | 58.90 | 0.12 | 0.02 | 0.04 |
| | RN18-NT ($\sigma = 0.1$) | 56.50 | 0.26 | 0.16 | 0.38 |
| TinyImagenet | RN18-AT-ES | 45.80 | 25.40 | **21.60** | 21.92 |
| | RN18-KAP-NT ($\sigma = 0.1, K \in \{3\}$) | 39.60 | **27.24** | 20.78 | **22.50** |
| | RN18-NT | 68.68 | 0.0 | 0.0 | 0.0 |
| | RN18-NT ($\sigma = 0.1$) | 51.20 | 0.19 | 0.20 | 0.0 |
| Imagenet | RN18-AT† | 53.20 | 19.23 | 6.00 | 6.00 |
| | RN18-KAP-NT ($\sigma = 0.1, K \in \{3\}$) | 9.60 | 9.14 | 3.00 | 4.80 |
| | RN18WideX4-AT† | 62.00 | 27.81 | 11.80 | 11.60 |
| | RN18WideX4-KAP-NT ($\sigma = 0.1, K \in \{3\}$) | 38.00 | **30.91** | **21.40** | **31.00** |

between 1 to 3 layers while keeping the network width fixed. We observed that increasing the network depth improves its robustness against AutoAttack for both KAP and non-KAP networks. However, this improvement was much more pronounced for KAP networks (Fig. A2a).

Since the number of convolutional and KAP layers were co-varying in the previous experiment, it is possible that the observed improvement in network robustness was solely due to the increasing network depth. To investigate this, in the second experiment, we kept the network depth fixed while changing the number of KAP layers. We trained the following three variations of a 3-layer CNNs: 1) one KAP layer after first convolutional layer; 2) one KAP layer after first and second convolutional layers; 3) one KAP layer after each of the three convolutional layers. In all 3 architectures, the KAP layer consisted of $3 \times 3$ kernel with stride 1. Activation noise with $\sigma = 0.1$ was added after each KAP layer. We validated the adversarial accuracy of each of these models with AutoAttack-$L_\infty$ with varying epsilon (Fig. A2b). We observed that while the number of convolutional layers was fixed, adding more KAP layers consistently improved the network robustness against AutoAttack.

In the third experiment, we varied the KAP kernel size between 1 to 4 to investigate the effect of larger ensemble sizes on the network robustness. In each network, we set the KAP stride equal to the KAP kernel size to avoid overlapping between ensembles, and also increased the network width by the same factor as the KAP kernel size to allow each network to learn the same number of kernel ensembles. We found that increasing the KAP kernel size and consequently the ensemble size further improves the network robustness against AutoAttack (Fig. A2c). The resulting kernel ensembles in each of these models are shown in Fig.A3.

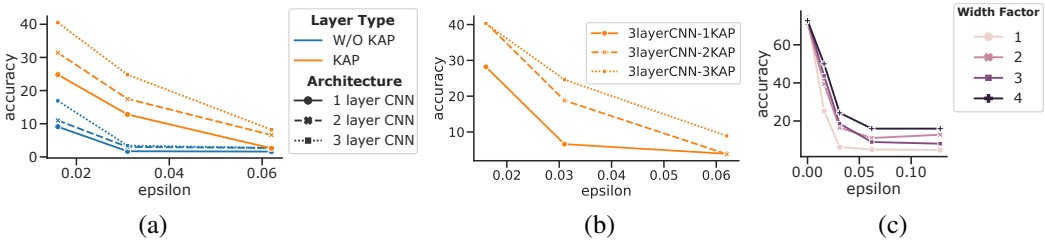

(a)          (b)          (c)

Figure A2: Effect of network depth and kernel ensemble size on the robustness to AutoAttack ($\epsilon = 0.031$) on CIFAR10 dataset. a) increasing the number of layers in a CNN from 1-3 significantly improves its robustness to AutoAttack in networks with KAP compared to networks without KAP; b) increasing the number of KAP layers while keeping the number of convolutional layers fixed (3) improves robustness to AutoAttack;c) increasing the kernel ensemble size by increasing the KAP kernel size (expansion) improves the network robust accuracy against AutoAttack. All attacks are performed on the corresponding model without noise.

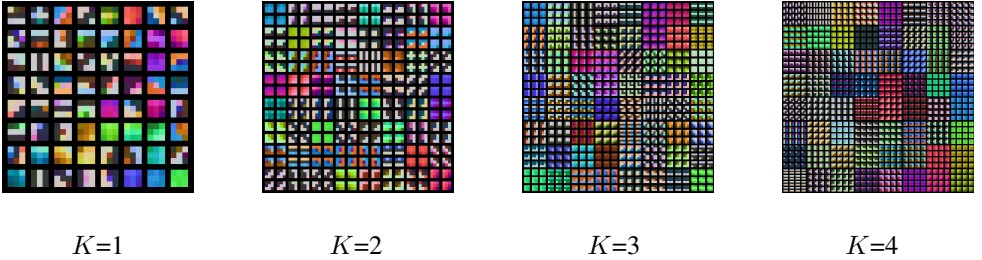

$K{=}1$          $K{=}2$          $K{=}3$          $K{=}4$

Figure A3: Visualization of the learned weights in the first layer of 3-layer convolutional networks with varying KAP kernel size $K$.

Table A5: Robust accuracy against transfer attacks on CIFAR10. Attacks were generated using each *Reference Model* and tested on the *Model*. For all attacks we used $\epsilon = 0.031$.

| MODEL | REFERENCE MODEL | AUTOATTACK |
|---|---|---|
| RN18-KAP-NT ($\sigma = 0.1, K = 3$) | RN18-NT | 76.4 |
| | RN18-AT | 66.19 |
| RN18-KAP-NT ($\sigma = 0.2, K = 3$) | RN18-NT | 68.6 |
| | RN18-AT | 61.84 |

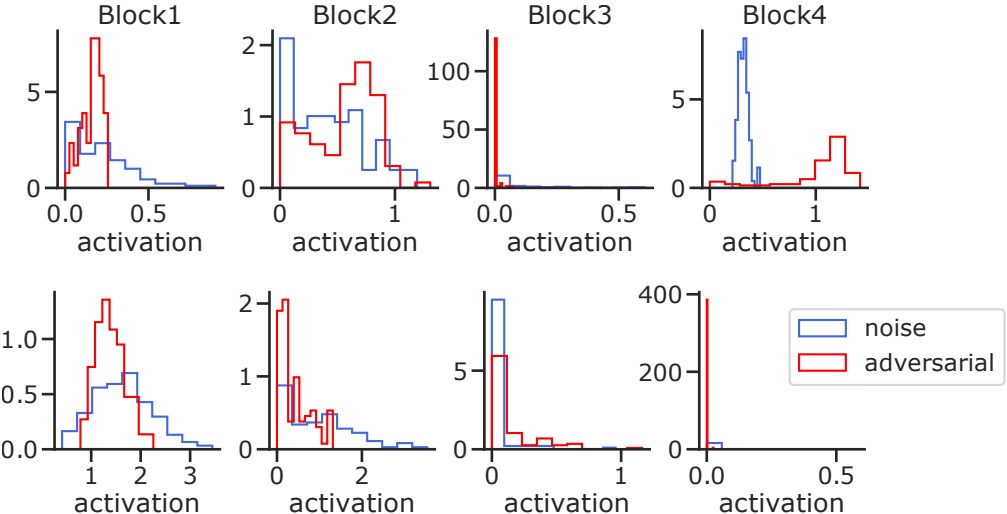

Figure A4: Distribution of layer activations in a random kernel for RN18-NT($\sigma = 0.1$) (top) and RN18-NT-KAP($\sigma = 0.1$) (bottom) trained on CIFAR10 dataset. We extracted the layer activations in response to a single random input image perturbed 100 times with random Gaussian noise or AutoAttack. Distributions of layer activations to noise and adversarial examples exhibit increased divergence in Block 4 in RN18-NT model but not in RN18-NT-KAP. The same pattern is replicated when considering other randomly selected input images from the validation set.

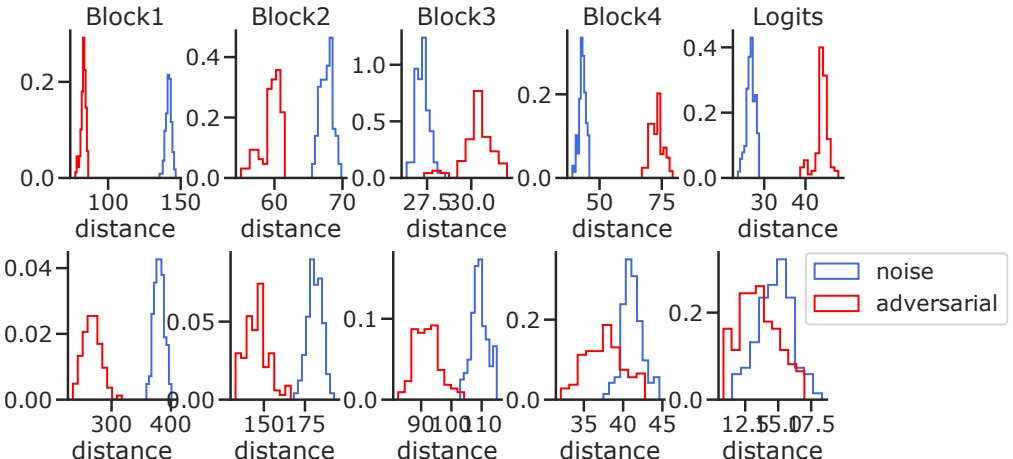

Figure A5: Distribution of perturbation magnitude in layer activations for RN18-NT($\sigma = 0.1$) (top) and RN18-NT-KAP($\sigma = 0.1$) (bottom) trained on CIFAR10 dataset. We extracted the layer activations in response to a single random input image perturbed 100 times with random Gaussian noise or AutoAttack. Perturbation magnitude is computed as the $L_2$ distance between perturbed and clean input activations ($f^{(l)}(x + n_i) - f^{(l)}(x)$ for noise and $f^{(l)}(x'_i) - f^{(l)}(x)$ for adversarial perturbations). Distributions of layer activations to noise and adversarial examples exhibits divergence in later blocks in RN18-NT model but not in RN18-NT-KAP.

Table A3: Attack hyperparameters used to validate model robustness on each dataset.

| Attack | Dataset | Steps | Size ($\epsilon$) | More |
|---|---|---|---|---|
| PGD-$L_\infty$ | CIFAR | 20 | $\frac{8}{255}$ | step=$\frac{2}{255}$ |
| | TinyImagenet | | $\frac{4}{255}$ | step=$\frac{2}{255}$ |
| | Imagenet | | $\frac{4}{255}$ | step=$\frac{1}{255}$ |
| $AutoAttack_{std}$ | CIFAR | 100 | $\frac{8}{255}$ | default standard AA - |
| | TinyImagenet | | $\frac{4}{255}$ | APGD-ce, APGD-t, FAB, |
| | Imagenet | | $\frac{4}{255}$ | SQUARE |
| $AutoAttack_{rnd}$ | CIFAR | 100 | $\frac{8}{255}$ | default random AA - |
| | TinyImagenet | | $\frac{4}{255}$ | APGD-ce and APGD-dlr, |
| | Imagenet | | $\frac{4}{255}$ | each with 20 EoT iterations |
| SQUARE | CIFAR | 5000 | $\frac{8}{255}$ | |
| | TinyImagenet | | $\frac{4}{255}$ | default SQUARE setting |
| | Imagenet | | $\frac{4}{255}$ | |

Table A4: Comparison of training speed between alternative models. All training times were computed on the CIFAR10 dataset and ResNet18 architecture using a single A100 GPU.

| Dataset | Model | Ave. Training Speed / Epoch |
|---|---|---|
| CIFAR10 | RN18-NT | $12.5 \pm 0.5$ sec |
| | RN18-AT | 100.75 sec |
| | RN18-KAP-NT ($\sigma = 0.1, K = 3$) | $14.5 \pm 0.5$ sec |

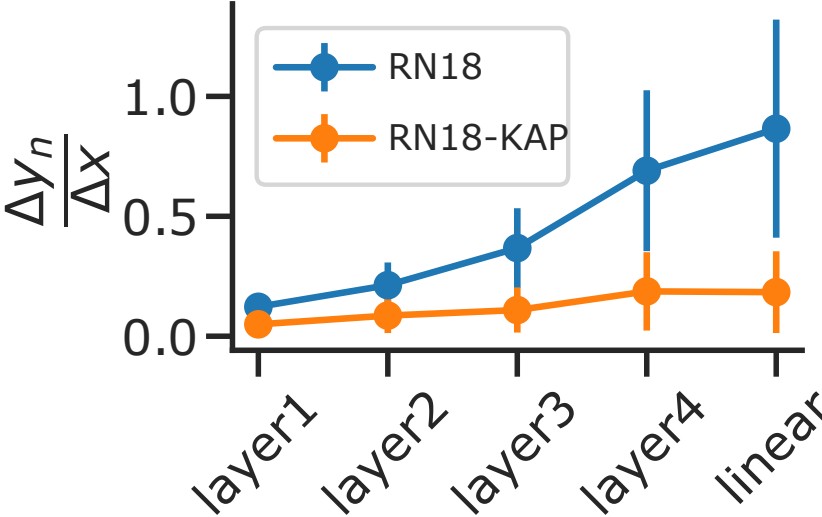

Figure A6: Normalized magnitude of change in each layer's activations in response to adversarial perturbations ($\frac{\|y_{adv} - y_{cln}\|}{\|y_{cln}\|}$) for AutoAttack $L2, \epsilon = 1$. on CIFAR10 dataset for RN18-NT ($\sigma = 0.1$) and RN18-KAP-NT ($\sigma = 0.1$).

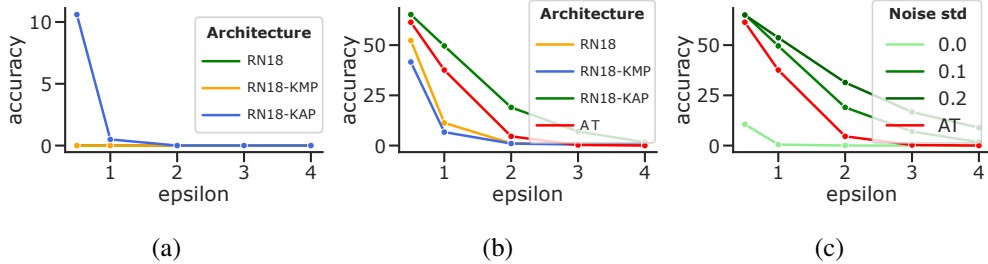

Figure A7: Robust accuracy in RN18-KAP model against AutoAttack-L2 with various attack strength $\epsilon$ on CIFAR10 dataset. (a) RN18 and RN18-KAP ($\sigma = 0$, $K = 3$); (b) RN18, RN18-KAP, RN18-KMP ($\sigma = 0.1$, $K = 3$) and AT; (c) RN18-KAP ($K = 3$) for various noise levels $\sigma$.

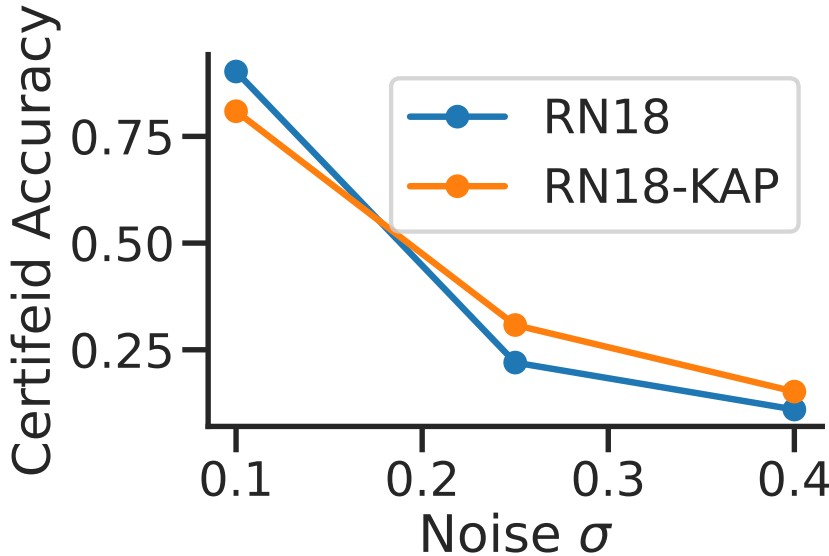

Figure A8: Certified accuracy of RN18-NT ($\sigma = 0.1$) and RN18-NT-KAP ($\sigma = 0.1$) models on 500 samples from CIFAR10 dataset for varying amount of noise levels $\sigma$. Certified accuracy was estimated using the Monte Carlo procedure from (Cohen et al., 2019) using 100 samples for selection and 100000 samples for estimation.

Table A6: Comparison of adversarial accuracy against AutoAttack-$L_\infty$ with $\epsilon = 0.031$ on CIFAR10 for variations of the models with and without activation and input noise.

| DATASET | MODEL | INPUT NOISE | ACTIVATION NOISE | AUTOATTACK$_{std}$ |
|---------|-------|-------------|------------------|---------------------|
| CIFAR10 | RN18-NT ($\sigma = 0.1$) | ✗ | ✗ | 5.00 |
| | | ✗ | ✔ | 8.90 |
| | | ✔ | ✗ | 16.9 |
| | | ✔ | ✔ | 21.10 |
| CIFAR10 | RN18-KAP-NT ($\sigma = 0.2, K \in \{3\}$) | ✗ | ✗ | 8.3 |
| | | ✗ | ✔ | 41.80 |
| | | ✔ | ✗ | 57.30 |
| | | ✔ | ✔ | 60.70 |

