# OpenReview forum: "Learning Robust Kernel Ensembles with Kernel Average Pooling"
_ICLR.cc/2023/Conference — Submitted to ICLR 2023_

### Official Review · Reviewer_6v33 · 2022-10-24

**Confidence:** 3
**Correctness:** 2
**Technical Novelty And Significance:** 3
**Empirical Novelty And Significance:** 3
**Recommendation:** 6

**Clarity, Quality, Novelty And Reproducibility:**

I think overall the writing is clear and the proposed technique, KAP, is novel.  I do have some concerns about the rigor of the experiments though (see weaknesses), especially since many randomized empirical defenses have been proposed in the past, but these have also been broken by stronger attacks (ie. Sitawarin et al. 2022).

Sitawarin, Chawin, Zachary J. Golan-Strieb, and David Wagner. "Demystifying the adversarial robustness of random transformation defenses." International Conference on Machine Learning. PMLR, 2022.

**Strength And Weaknesses:**

Strengths:
- writing is clear and easy to understand
- diagrams help with understanding KAP
- experiments on a variety of datasets

Weaknesses:
- questionable robustness- Since this work proposes a randomized defense (without proposing or testing any adaptive attack against this defense) it is unclear whether KAP really provides robustness or not.
- lack of ablation- the work immediately jumps into using multiple KAP layers with noise added before every layer.  What happens if you only add noise to the input?  Would using KAP improve on certified robustness via randomized smoothing in this case?  What about changing the number of KAP layers vs regular convolutional layers?
- For CIFAR-10, I think the robust accuracy numbers for AT are a little low, probably due to suboptimal hyperparameters.  Could the authors use the training setup from Gowal et al. 2020?  I think that the choice in learning rate scheduler seems to change robustness numbers of AT by a lot.
Gowal, Sven, et al. "Uncovering the limits of adversarial training against norm-bounded adversarial examples." arXiv preprint arXiv:2010.03593 (2020).
- Since TRADES is also commonly used, it would also be good to use TRADES as a baseline too (using the training setup from Gowal et al 2020)
- It would also be interesting to provide results on L2 robustness as well; after all the motivation of KAP was based on randomized smoothing which is a defense against L2 attacks.

**Summary Of The Paper:**

This paper proposes an architectural component called kernel average pooling which averages kernels (channels) of the input to the kernel average pooling layer.  They show that when using multiple KAP layers combined with random noise added to the input of each KAP layer, the adversarial robustness of the final model increases.

**Summary Of The Review:**

Overall, I think that the authors should try using adaptive attacks against KAP and perform more ablations to better understand the impact of the noise in KAP and the impact of KAP layers on robustness.

---

> ### Author Response · Authors · 2022-11-15
> **The criticism of robustness evaluations is unspecific and dismissive of our best efforts**
>
> We thank the reviewer for providing feedback on our submission. The reviewer has 1) questioned the evaluations of robustness; 2) criticized the lack of ablation experiments. In response, we argue that 1) the reviewer’s criticism of our evaluations is unspecific and only broadly mentions “adaptive attacks” without determining what they mean by it. This comment also comes despite our best efforts in anticipating such possibilities and taking measures to rule them out (detailed below); 2) the ablation experiments investigating the effect of number of layers and distinguishing the role of each component (KAP and activation noise) were already included in our original submission but were possibly overlooked by the reviewer. We believe our responses should address the reviewer’s concerns and welcome any additional discussions.
>
> **1) Robustness Evaluations**
>
> - *“questionable robustness- Since this work proposes a randomized defense (without proposing or testing any adaptive attack against this defense) it is unclear whether KAP really provides robustness or not.” / “I think that the authors should try using adaptive attacks against KAP …”*
>
> The reviewer mentions that because of the randomness in our proposal, it is likely that our evaluations of robustness are not sufficient. In our evaluations, we paid special attention to this possibility and took several measures to make sure the stochasticity alone is not the sole source of robustness. In particular, 1) all attack models were applied on the model with no activation noise, to make sure that the stochasticity does not interfere with the gradient estimation; 2) All of our main evaluations of robustness were done using AutoAttack that is a standard attack model in the literature and consists of an ensemble of four attacks including the SQUARE black box attack; 3) in cases where the attacker was applied on the model with stochasticity (included in Tables A1 and A2 of the appendix), we additionally performed the random version of AutoAttack with 20 EoT to address this concern. We believe these evaluations already indicate reliable robustness against a wide range of strong adversarial robustness benchmarks. Beyond these measures, and with lack of specificity in the reviewer’s comments, we are unsure what the reviewer means precisely by “using adaptive attacks” and would surely welcome any further suggestions by the reviewer.
>
> - *"For CIFAR-10, I think the robust accuracy numbers for AT are a little low, probably due to suboptimal hyperparameters. Could the authors use the training setup from Gowal et al. 2020? I think that the choice in learning rate scheduler seems to change robustness numbers of AT by a lot. Gowal, Sven, et al. "Uncovering the limits of adversarial training against norm-bounded adversarial examples." arXiv preprint arXiv:2010.03593 (2020)."*
>
> We believe that the reviewer is misremembering the results from Gowal et al. paper as all results in this paper are presented using the Wide Resnet architecture (WRN-28-10, WRN-34-20, WRN-70-16) and not ResNet18 which is the architecture used in our analyses.
>
>
> - *"Since TRADES is also commonly used, it would also be good to use TRADES as a baseline too (using the training setup from Gowal et al 2020)"*
>
> We believe that the addition of TRADES as a new baseline does not in any way affect the validity of our claims or the conclusions and is thus unnecessary. Our reasoning for this argument is twofold:
>
> 1) Our statement of claim in the paper clearly contrasts our proposed KAP model with adversarial training. We do not claim a new state of the art and because of that reason our evaluations only consider adv. training and other possible relevant baselines like models trained on noise.
>
> 2) Two recent papers Rice et al. 2020 and Gowal et al. 2020 have shown that adversarial training with early stopping performs almost identical to TRADES.
>
> - *"It would also be interesting to provide results on L2 robustness as well; after all the motivation of KAP was based on randomized smoothing which is a defense against L2 attacks."*
>
> To address this comment, we performed new evaluations using AutoAttack with L2 norm on our KAP models, the baseline RN18 with and without activation noise as well as a new adversarially trained RN18 baseline model trained with L2 ($\epsilon$=1.) PGD. Both KAP-RN18-$\sigma$=0.1 and -$\sigma$=0.2 were consistently more robust than the L2-AT model across different epsilon values (0.5-4). The clean accuracy of the AT-L2 baseline was however again slightly better than the KAP-RN18-$\sigma$=0.1 model (81.07% vs 79.09%). These results were included as a new Fig. A7 in the appendix.
>
>
> [continued in next comment]

---

> > ### Author Response · Authors · 2022-11-15
> > **The criticism of robustness evaluations is unspecific and dismissive of our best efforts (Cont'd)**
> >
> > [Continued from previous comment]
> >
> > - *"I do have some concerns about the rigor of the experiments though (see weaknesses), especially since many randomized empirical defenses have been proposed in the past, but these have also been broken by stronger attacks (ie. Sitawarin et al. 2022). Sitawarin, Chawin, Zachary J. Golan-Strieb, and David Wagner. "Demystifying the adversarial robustness of random transformation defenses." International Conference on Machine Learning. PMLR, 2022."*
> >
> > The referenced paper investigates the use of defenses where non-differentiable transformations are used in the model and impede gradient estimation. This point is not relevant to our proposed approach because of two reasons. 1) all the results presented in the paper are from applying the attacker on the model with no activation-noise and stochasticity. This evaluation procedure was chosen to make sure the stochasticity does not interfere with the attacker. 2) the only transformation which we have used in our model is additive Gaussian noise which is implemented in a  fully differentiable manner using the PyTorch library.
> >
> > **2) Lack of Ablations**
> >
> > - *“lack of ablation- the work immediately jumps into using multiple KAP layers with noise added before every layer. What happens if you only add noise to the input? Would using KAP improve on certified robustness via randomized smoothing in this case? What about changing the number of KAP layers vs regular convolutional layers?” / “... perform more ablations to better understand the impact of the noise in KAP and the impact of KAP layers on robustness”*
> >
> > *Effect of number of layers*: The proposed experiment on investigating the effect of depth was already performed in our original submission and the results were included in the appendix (see figure A2a). We chose to put these results in the appendix because of the constraints in the number of pages in ICLR submission. We encourage the reviewer to revisit our manuscript and in particular to view section A3 in the appendix where we demonstrate the varied the number of layers in a CNN from 1 to 3 layers and show that increasing depth has a strong positive effect on robustness in KAP models. These experiments were referenced in Results on the last paragraph of section 4.2.
> >
> > *Ablation of noise and KAP*: We believe that the reviewer has overlooked our baseline models that were already included in our original submission. In our original submission, we had included the following baseline models: a) model with no KAP and no activation noise; b) model with no KAP and activation noise; c) model with KAP and no activation noise;  and d) model with KAP and activation noise (Please see Tables 1 and 2 and Figure 2a). We expected that these baselines would be sufficient for making the deduction about the contributions of each of the components of our approach on improving robustness. We welcome any additional suggestions from the reviewer in that regard.
> >
> > *Input-only noise model*: We appreciate the additional proposed analyses. To address this comment, we performed several additional analyses on the RN18 and RN18-KAP models on CIFAR10. In particular we performed evaluation of a) RN18 with added input noise at test time; b) RN18-KAP with only input noise at test time; c) RN18 with added noise to input and activation at test time; d) RN18-KAP with added input noise and activation at test time. Please note that all of our results included in the original manuscript only used activation noise and no noise was added to the input at test time. From the newly performed analysis we found that 1) additive input noise had a more pronounced effect on robustness to AutoAttack compared to activation noise in the intermediate layers; 2) combining input noise and activation noise further boosted robustness against AutoAttack. These results are added in Table-A6 in the appendix.

---

> > > ### Comment · Reviewer_6v33 · 2022-11-17
> > > **Thank you for the response**
> > >
> > > Thank you for the clarifications and additional experiments.  Please see additional comments below:
> > >
> > > **Robustness Evaluations**
> > >
> > > *Rigor of evaluations:*
> > > Thank you for the clarification.  I am a little confused by the wording that "attacks are applied on the model with no activation noise".  Do you mean that you generate the attacks on the model (same weights and architecture) with no noise, and then use these attacks on the model with noise?  If that's the case, could the wording used in the text be changed to "attacks are generated on the corresponding model without noise" for clarity?
> > >
> > > Also to confirm, in 20 EoT, the 20 refers to the number of samples used to estimate the gradients right?  Since you are using activation noise after every convolutional layer in ResNet-18 (which I think has 18 convolutional layers), I do not think that 20 samples is enough to estimate gradients accurately (which is probably why the EoT ends up being a weaker attack compared to the ones used in the main text.
> > >
> > > *Baselines:* Yes, I am aware that Gowal et al. 2020 experiments with only WRN, I was referencing the paper to encourage the authors to use the same learning rate scheduling.  In the past, I have trained ResNet-18 models using the lr scheduling and obtained ~47% robust accuracy for CIFAR-10 on AutoAttack, which is why I think that the 43% reported in the paper for AT is a little low.  Using TRADES too can achieve ~48% robustness with ResNet-18.  Even though the authors are not claiming state-of-the-art, it is still important to have accurate baselines for AT.
> > >
> > > **2) Ablations**
> > >
> > > *Effect of number of layers* I apologize for the lack of clarity in my original review.  What I meant was an ablation on the number of KAP layers.  Currently, if I am understanding the experimental setup correctly, every convolutional layer is followed by a KAP layer.  But it is unclear whether that is necessary or not.  What if we only have KAP after the 1st convolution? Or after the first 2 convolutional layers?  Does increasing the number of KAP layers strictly increase robustness?
> > >
> > > *Ablation of noise and KAP* Thank you for the clarification, I think I misinterpreted Figure 2.  Could the authors update the legends in Figure 2 (and similar figures in the Appendix) so that it matches the caption?  Does "mean" mean the KAP model?  What are "max" and "None"?
> > >
> > > *Input-only noise model* Thank you for the additional experiments.  Since the motivation of KAP was based on randomized smoothing, I would still like to see some comparisons of certified robustness via randomized smoothing between an architecture using KAP and without KAP.

---

> > > > ### Author Response · Authors · 2022-11-18
> > > > **New evaluations further support our claims**
> > > >
> > > > We thank the reviewer for additional comments. The reviewer requested new experiments to further assess the reliability of our robustness evaluations and suggested changes to the text and figures to improve clarity (see our detailed responses below). To address these remaining points, we performed several new experiments that showed: 1) increasing the number of KAP layers without increasing network depth has a positive effect on network robustness; 2) KAP networks have higher certified robustness compared to the baseline model; 3) increasing the number of samples for estimating gradients for attacks applied on models with activation noise in some cases reduced the robust accuracy. But importantly, this change did not affect any of our main results presented in the paper where all attacks were applied on the corresponding models without noise. We believe that these new results provide further support for our claims and verify the reliability of our robustness evaluations and welcome any additional discussions.
> > > >
> > > > **1) Robustness Evaluations**
> > > >
> > > > - *"Rigor of evaluations: Thank you for the clarification. I am a little confused by the wording that "attacks are applied on the model with no activation noise". Do you mean that you generate the attacks on the model (same weights and architecture) with no noise, and then use these attacks on the model with noise? If that's the case, could the wording used in the text be changed to "attacks are generated on the corresponding model without noise" for clarity?"*
> > > >
> > > > The reviewer has correctly understood our evaluation setup. To reiterate, all results except those in Tables A1 and A2 were obtained by applying adversarial attacks on instances of each model without noise (input or activation). To further clarify this point, we incorporated the reviewer’s suggested wording and revised the text explaining this procedure.
> > > >
> > > > - *"Also to confirm, in 20 EoT, the 20 refers to the number of samples used to estimate the gradients right? Since you are using activation noise after every convolutional layer in ResNet-18 (which I think has 18 convolutional layers), I do not think that 20 samples is enough to estimate gradients accurately (which is probably why the EoT ends up being a weaker attack compared to the ones used in the main text."*
> > > >
> > > > We thank the reviewer for the insight. We confirm that in our original evaluations using $AutoAttack_{rnd}$, we had used 20 samples to estimate the gradients (i.e. EoT=20). To address this concern, we additionally evaluated the models with stochastic components in Table A1 with $AutoAttack_{rnd}$ and EoT=50 attack. We observed that the attack with larger EoT, in some cases yielded lower accuracy than those obtained with EoT=20 (e.g. 56.1% vs. 61.64% for RN18-KAP-($\sigma=0.1$)). However, we emphasize that our main results presented in the main body (Tables 1-2) remain unaffected by this as they were obtained by applying attackers on “the corresponding model without noise” and thus are not expected to require additional sampling for estimating the true gradients. For transparency, we additionally included the results with EoT=50 in our supplementary table A1.
> > > >
> > > > - *"Baselines: Yes, I am aware that Gowal et al. 2020 experiments with only WRN, I was referencing the paper to encourage the authors to use the same learning rate scheduling. In the past, I have trained ResNet-18 models using the lr scheduling and obtained ~47% robust accuracy for CIFAR-10 on AutoAttack, which is why I think that the 43% reported in the paper for AT is a little low. Using TRADES too can achieve ~48% robustness with ResNet-18. Even though the authors are not claiming state-of-the-art, it is still important to have accurate baselines for AT."*
> > > >
> > > > Thank you for the clarification. We understand that additional hyperparameter optimizations like learning rate schedule may enable further improvements in some of our baseline models. However, 1) we did not perform any similar hyperparameter optimization for any of our proposed models and simply used the standard training procedure from the literature; 2) our methodology for training the AT baseline with early stopping follows the Rice et al 2020 paper that has shown similar results using the ResNet18 architecture. To ameliorate the reviewer’s concern and to clarify the exact methodology used for training our baseline AT model, we added the reference to the Rice et al. 2020 paper both in our description of adversarial training procedure as well as in Tables 1-2.
> > > >
> > > > Rice, Leslie, Eric Wong, and Zico Kolter. "Overfitting in adversarially robust deep learning." International Conference on Machine Learning. PMLR, 2020.
> > > >
> > > > [continued in next comment]

---

> > > > > ### Author Response · Authors · 2022-11-18
> > > > > **New evaluations further support our claims (Cont'd)**
> > > > >
> > > > > [Continued from previous comment]
> > > > >
> > > > > **2) Ablations**
> > > > >
> > > > > - *"Effect of number of layers I apologize for the lack of clarity in my original review. What I meant was an ablation on the number of KAP layers. Currently, if I am understanding the experimental setup correctly, every convolutional layer is followed by a KAP layer. But it is unclear whether that is necessary or not. What if we only have KAP after the 1st convolution? Or after the first 2 convolutional layers? Does increasing the number of KAP layers strictly increase robustness?"*
> > > > >
> > > > > Thank you for this clarification. To address the reviewer’s concern, we trained three 3-layer convolutional networks (similar to the one presented in Figure A2 (a) as “tripleconv” now renamed to "3layer CNN"). These three networks only differed in the position and number of KAP layers in their architecture as follows: 1) one KAP layer after first convolutional layer; 2) one KAP layer after first and second convolutional layers; 3) one KAP layer after each of the three convolutional layers. In all 3 architectures, the KAP layer consisted of 3x3 kernel with stride 1. Activation noise with $\sigma=0.1$ was added after each KAP layer. We validated the adversarial accuracy of each of these models with AutoAttack-$L_\infty$ with varying epsilon and included these results as a new subfigure in Figure A2. Briefly, we observed that within the architecture with same number of convolutional layers (3), adding more KAP layers consistently improved the network robustness against AutoAttack. This new result further supports the importance of KAP layers for robustness and highlights its effect in a manner that is disentangled from the network depth.
> > > > >
> > > > > - *"Ablation of noise and KAP Thank you for the clarification, I think I misinterpreted Figure 2. Could the authors update the legends in Figure 2 (and similar figures in the Appendix) so that it matches the caption? Does "mean" mean the KAP model? What are "max" and "None"?"*
> > > > >
> > > > > We thank the reviewer for the additional suggestion. To improve clarity, we edited the figure legends to match those in their corresponding captions. To clarify, “mean” refers to a model with Kernel Average Pools (RN18-NT-KAP), “None” refers to a model without KAP (RN18-NT), and “max” refers to a model with Kernel Max Pooling (RN18-NT-KMP).
> > > > >
> > > > > - *"Input-only noise model Thank you for the additional experiments. Since the motivation of KAP was based on randomized smoothing, I would still like to see some comparisons of certified robustness via randomized smoothing between an architecture using KAP and without KAP."*
> > > > >
> > > > > To address this concern, we empirically evaluated the robustness of RN18-NT($\sigma=0.1$) and RN18-KAP-NT($\sigma=0.1$) models on CIFAR10 dataset using Cohen et al. 2019 official code (http://github.com/locuslab/smoothing). We adopted their default configuration for CIFAR10 experiments by including every 20th sample from the test set in the analysis (total of 500 samples), 100,000 evaluation samples and 100 selection samples. We observed that the RN18-KAP-NT($\sigma=0.1$) showed higher certified accuracy compared to RN18-NT($\sigma=0.1$) on larger sigma values (e.g. 31% vs. 22% on $\sigma=0.25$). The relatively lower certified accuracy for lower $\sigma$ values is most likely due to lower clean accuracy of the KAP model. We added these results as a new supplementary figure in the appendix (Fig. A8).

---

> > > > > > ### Comment · Reviewer_6v33 · 2022-12-07
> > > > > > **Thank you for the response**
> > > > > >
> > > > > > Thank you for the additional experiments and clarifications.  I believe most of my concerns are addressed and will raise my score to a 6.

---

> > > > > > > ### Author Response · Authors · 2022-12-07
> > > > > > > **Thank you!**
> > > > > > >
> > > > > > > We are happy that we were able to address most of the reviewers concerns and we thank the reviewer for considering to raise their score.

---

### Official Review · Reviewer_is9m · 2022-10-24

**Confidence:** 4
**Correctness:** 3
**Technical Novelty And Significance:** 3
**Empirical Novelty And Significance:** 3
**Recommendation:** 3

**Clarity, Quality, Novelty And Reproducibility:**

- This paper has problems with clarity. The description of the algorithm is difficult to understand. The notations in Algorithm 1 such as Vec, Vec^-1, Pad, AvePool are not properly defined.
- Noise is used in the proposed algorithm (experiments in Section 4) but the authors have not been properly described how they are used in the algorithm in Section 3
- The clearest explanation of kernel average pooling is in Appendix A1; the descriptions in Section 3.2 and Algorithm 1 are not helpful compared to Figure A1. The authors should consider revising that section by including the figure.


**Strength And Weaknesses:**

Strengths:
- The authors combine averaging kernels with additive noise to improve the robustness of neural networks against adversarial attacks; some of these ingredients are known to help improve robustness.

- The method is competitive with state-of-art methods based on adversarial training on some datasets, without the use of adversarial training.

Weaknesses:
- Although there are some good empirical results, it is hard to understand and verify why the method works. Is it due to the use of noise for regularization in the network, or the use of averaging of kernels, or both? There is no ablation studies on these factors that might contribute to the robustness of the model, making the current study difficult to understand and reproduce.

- The authors described 2D kernel averaging in Appendix A1. Yet unlike the spatial dimensions, there is no topological structure relating the different kernel dimensions. Why choose 2d instead of 3d or 1d?

- While noise is known to improve robustness in previous work, it is not clear why kernel averaging helps improve robustness. The authors should consider further empirical/theoretical analysis to illustrate why it works, such as looking at its smoothing properties or effects on Lipschitz constants of the network.


**Summary Of The Paper:**

The authors propose an algorithm to improve the adversarial robustness of neural networks based on kernel averaging and noise. They show that their method is competitive with adversarial training in some cases. However, the algorithm description is not clear enough and there are also not sufficient investigations on why the method works.


**Summary Of The Review:**

The authors present a method for improving the robustness of neural networks based on kernel averaging and additive noise. Although there are some encouraging empirical results, it is difficult to understand why the method works. The authors should improve the presentation of the method, especially description of the algorithm, to improve the readability of the paper. The authors should also consider including ablation studies to help the readers understand what makes the method work.

---

> ### Author Response · Authors · 2022-11-15
> **The requested ablation studies were presented as baselines**
>
> We thank the reviewer for providing feedback on our submission. The reviewer criticizes 1) lack of ablation experiments; 2) lack of explanation for why the method works. We argue that 1) all the baseline models constituting the ablation experiments that we believe the reviewer is asking for were already included in our original submission; 2) We have provided detailed justification for why our proposed approach improves robustness and the role of each of its components in section 3.4. Below we provide detailed responses to each of the reviewer’s comments. We believe these responses clarify points raised by the reviewer and we welcome any further points the reviewer may want to additionally discuss.
>
> **1) Lack of Ablations**
>
> - *"Although there are some good empirical results, it is hard to understand and verify why the method works. Is it due to the use of noise for regularization in the network, or the use of averaging of kernels, or both? There is no ablation studies on these factors that might contribute to the robustness of the model, making the current study difficult to understand and reproduce."*
>
> *Lack of Ablations*: The reviewer must have missed some of our key results included in the main text and the appendix. In our original submission, we had already included four baseline models that we believe are the exact “ablation studies” that the reviewer is asking for. These baselines aimed at disentangling the effect of KAP and noise on robustness (as the reviewer has also requested) and showcase their complementary roles in improving network robustness. The baselines include KAP model without noise (Fig. 2a), model without KAP and activation noise (Fig. 2a and Table 1: RN18-NT), and model without KAP and with activation noise (Table 1-RN18-NT(\sigma=0.1). We believe these baselines should have already demonstrated the role of each of these components on network robustness. We would appreciate it if the reviewers could tell us what other ablations they want to examine.
>
> *Reproducibility*: We had already provided the code for our experiments in the supplementary material that can be readily used to reproduce all of our results on CIFAR10 and CIFAR100 datasets. We would appreciate it if the reviewer could inform us what other form of reproducibility they are expecting.
>
> **2) Method clarity**
>
> - *"The authors described 2D kernel averaging in Appendix A1. Yet unlike the spatial dimensions, there is no topological structure relating the different kernel dimensions. Why choose 2d instead of 3d or 1d?"*
>
> As we had already discussed in our original submission (page 5-second paragraph), “KAP could more generally be applied on any D- dimensional tensor arrangement of kernels”. This means that the number of dimensions is a design parameter that could be set by the experimenter. In our empirical experiments, we chose a 2D KAP to make the number of dimensions consistent with the rest of the operations in the network performed on the spatial dimensions (all 2D) and for easier visualization of the learned kernels.
>
> - *"While noise is known to improve robustness in previous work, it is not clear why kernel averaging helps improve robustness. The authors should consider further empirical/theoretical analysis to illustrate why it works, such as looking at its smoothing properties or effects on Lipschitz constants of the network."*
>
> In section 3.4, we have discussed in length why kernel averaging results in improved robustness. Briefly, the implications of our proposed method for robustness can be understood through randomized smoothing (Lecuyer et al. 2019 and Cohen et al. 2019) and more specifically a recent variation of it (Horvath et al 2022) which extended it to an ensemble of models. As we showed in section 3.3, including KAP layers leads to learning ensembles of kernels (groups of kernels with similar selectivity). These kernel ensembles when combined with activation noise can be thought of as a randomized smoothing process at each layer. In this way, having several layers of KAP and activation noise amounts to a recursive form of randomized smoothing with ensembles similar to that in Horvath et al. 2022.
>
> To further accommodate the reviewer’s request, we additionally investigated the Lipschitz constants in the network with and without KAP layers. To do this, we quantified the norm of change in each layer’s activation due to adversarial perturbations ($\frac{||y_{adv}-y_{cln}||}{||y_{cln}||}$) of the input which could be thought of as an estimate of the worst case inputs that generate maximal change in the network output. We compared these values across different layers of RN18 and RN18-KAP models and show that the amount of change is consistently lower in layers of RN18-KAP compared to RN18 (see new supplementary figure in the appendix Fig. A6).
>
> [continued in next comment]

---

> > ### Author Response · Authors · 2022-11-15
> > **The requested ablation studies were presented as baselines (Cont'd)**
> >
> > [Continued from previous comment]
> >
> > - *"This paper has problems with clarity. The description of the algorithm is difficult to understand. The notations in Algorithm 1 such as Vec, Vec^-1, Pad, AvePool are not properly defined."*
> >
> > The reviewer must have missed the definition of these operators at the beginning of Algorithm 1. Briefly, $Vec$ is the vectorization operator commonly used in linear algebra. The $Vec^{-1}$ is the inverse of this operator that reshapes a vector into a multi-dimensional tensor. $Pad$ is the zero padding operator similar to the padding function used in spatial pooling and convolutional layers that add padding to the tensor to control for the output dimensions. The $AvePool$ is the average pooling function similar to those used in CNNs. For additional clarity and due to lack of space, we had included a figure in the appendix that visually demonstrates the steps summarized in Alg1.
> >
> > - *"Noise is used in the proposed algorithm (experiments in Section 4) but the authors have not been properly described how they are used in the algorithm in Section 3."*
> >
> > The reviewer may have misunderstood our approach as “activation noise” and “Kernel Average Pooling” operation are conceptually independent components in our proposal. As such, Algorithm 1 in section 3 defines the KAP operation which on its own does not include a noise component.  The addition of noise and its effect is explained in section 3.4 where we discuss how KAP and activation noise contribute to model robustness (e.g. Eq. 9). We invite the reviewer to revisit our manuscript and let us know how we could further improve the clarity of our manuscript to prevent future readers from being similarly confused.
> >
> > - *"The clearest explanation of kernel average pooling is in Appendix A1; the descriptions in Section 3.2 and Algorithm 1 are not helpful compared to Figure A1. The authors should consider revising that section by including the figure."*
> >
> > We thank the reviewer for this suggestion and feedback. The reason for having that figure in the appendix was strictly for keeping the length of our submission within the ICLR guidelines. We will consider reformatting the manuscript to make room for the addition of this figure in the main paper.

---

> > > ### Comment · Reviewer_is9m · 2022-12-09
> > > **After rebuttal**
> > >
> > > Thanks to the clarifications from the authors, but I think this paper still have major clarity issues.
> > >
> > > 1. Effect of noise, kernel averaging: Figure 2 and Table 1 contains some investigations into the contributions of noise level and kernel averaging. But if these are key results, they should be discussed in the main text and not just left in the figures and tables alone for the readers to draw conclusions from. Table 1 is also confusing. What does the author mean when they say all the attacks are performed without input or activation noise, when the KAP models are trained with noise (sigma=0.1, 0.2)? Shouldn't Expectation-over-Transformation (EoT) type of attacks be used for these randomized models (e.g., AutoAttack has a mode for attacking randomized models)?
> > >
> > > 2. Clarity of definitions: The definitions on top of Algorithm 1 are not proper definitions. What is the 'vectorization' operation used commonly in linear algebra? Vectorization usually means speeding up a loop by turning it into vector operations in computer science. Do you mean 'reshaping' a vector or tensor into different dimensions? If that is the case, what are the dimensions of the input x, and why do we need to take it as transpose x^T? And for 'Pad', what target dimensions do we want to pad the tensor to? Some mathematical rigor goes a long way here.
> > >
> > > Due to these issues I am still keeping my original score.

---

> > > > ### Author Response · Authors · 2022-12-09
> > > > **Further clarifications**
> > > >
> > > > We thank the reviewer for the follow up questions. The reviewer **1)** claims lack of discussion of some of our key results presented in Figure 2 and Table 1. We strongly disagree with this statement and provide detailed response below as to where each of those results were discussed in the main body of the paper in our original submission; **2)** criticizes the clarity of the definitions used in algorithm 1. We provide more clarifications on each of those definitions and suggest possible rewording to improve clarity on these definitions. We believe these clarifications should resolve all of reviewer’s concerns but remain open to further suggestions by the reviewer. Our detailed responses are below.
> > > >
> > > > - *”Figure 2 and Table 1 contains some investigations into the contributions of noise level and kernel averaging. But if these are key results, they should be discussed in the main text and not just left in the figures and tables alone for the readers to draw conclusions from.”*
> > > >
> > > > All results included in the main portion of the paper are key results and they are *all* throughly discussed in the main body of the manuscript. We believe that the reviewer has overlooked our detailed descriptions of these results in their review. We invite the reviewer to revisit a) last paragraph on page 7 of our manuscript where we have discussed each of our observations from Table 1 including the baselines and their contrasts to our proposed model; b) the second paragraph on page 8 of our manuscript where we have described each of the results in Fig 2 in detail. Beyond these numerical reports and verbal discussions, we are unsure what other form of discussions the reviewer is expecting. We remain open to any further suggestions.
> > > >
> > > > - *”What does the author mean when they say all the attacks are performed without input or activation noise, when the KAP models are trained with noise? Shouldn't Expectation-over-Transformation (EoT) type of attacks be used for these randomized models (e.g., AutoAttack has a mode for attacking randomized models)?”*
> > > >
> > > > As explained in each Table's caption, the adversarial attacks (i.e. examples) were generated on each model with no noise (same weights and architecture but with $\sigma=0$). These images were then fed to the model with its corresponding original noise parameter (i.e. nonzero \sigma). This procedure was performed to make sure the model's stochasticity does not interfere with any of the gradient estimation. Furthermore, because no stochasticity existed during the attack procedures, we did not use EoT in our results reported in Table 1-2. However, we also considered applying the attacks on full models with stochasticity where we additionally used the *rand* variation of AutoAttack (which includes 20 EoT). These results were reported in the appendix A.2 (Tables A1 and A2) and referred to in the main text.
> > > >
> > > > Finally, this point was also discussed with reviewer **6v33**. Following that reviewer’s suggestion, we changed the wording to the following to improve clarity: “All attacks are performed on the corresponding model without input or activation noise.”. We remain open to further suggestions.
> > > >
> > > > - *”What is the 'vectorization' operation used commonly in linear algebra? Vectorization usually means speeding up a loop by turning it into vector operations in computer science. Do you mean 'reshaping' a vector or tensor into different dimensions?”*
> > > >
> > > > The vectorization operator that we referred to in Alg. 1 converts a matrix (or generally a tensor) into a column vector. For reference, the definition of the vectorization operator in mathematics and linear algebra can be found here: https://en.wikipedia.org/wiki/Vectorization_(mathematics). To further improve clarity, we will add a more detailed description of this operation in text and add a reference to the wikipedia page to remove any possible uncertainties.
> > > >
> > > > - *”If that is the case, what are the dimensions of the input x, and why do we need to take it as transpose x^T?”*
> > > >
> > > > We thank the reviewer for brining this point to our attention. The transpose operator is applied on $Vec(x)$ and not $x$. In its revised form, the first line of Algorithm 1 should read as: $h \leftarrow Vec^{-1}_{\sqrt(D), \sqrt(D), WH}(Vec(x)^T)$
> > > >
> > > > - *”And for 'Pad', what target dimensions do we want to pad the tensor to? Some mathematical rigor goes a long way here.”*
> > > >
> > > > Like the reviewer, we also value rigor in our mathematical descriptions. However, we are unaware of any common notation for the *Padding* operator and its absence we used an informal notation following common practice in convolutional networks that pads the first two dimensions of a tensor (usually corresponding to W and H dimensions but here corresponding to the two kernel dimensions which appear first in the reshaped tensor). We will add more textual description of the dimensions over which padding is applied. We welcome any suggestions from the reviewer how to revise the notation for improved mathematical rigor.

---

### Official Review · Reviewer_ckSs · 2022-10-24

**Confidence:** 3
**Correctness:** 3
**Technical Novelty And Significance:** 3
**Empirical Novelty And Significance:** 3
**Recommendation:** 6

**Clarity, Quality, Novelty And Reproducibility:**

the paper is well written. The method is carefully described and sufficient information is provided to reproduce results.

**Strength And Weaknesses:**

Strength:
- The paper is well-written and easy to follow.
- The proposed method is technically sound and well motivated.
- The experiments are carefully conducted.
- The method seems to provide reasonable improvement in terms of adversarial robustness.

Weaknesses:
- Overall, I find this to be an interesting approach. The main complaint I have regarding this approach is that the robustness comes at the cost of significantly lower clean data accuracy, and at the same time, the improvement of adversarial robustness is not consistent, as seen from the fact that regular ResNet18 model with activation noise can sometimes outperform the proposed method in terms of adversarial robustness.
- While there are some insights offered on why the proposed method can help adversarial robustness in section 3.4, most of the explanation is quite descriptive and lacks concrete details in my opinion. It might be good to go in-depth on this and offer some additional insights mathematically.
- Maybe it makes sense to also benchmark the proposed method against some other cheap ensembling methods, such as [1]?
- Other than adversarial robustness, does the proposed method offer other benefits? Does it help combat robustness against natural noise, model calibration, etc., since these are common benefits observed from a regular neural network ensemble? It might be interesting to investigate further other potential benefits as well.

[1] Wen, Yeming, Dustin Tran, and Jimmy Ba. "Batchensemble: an alternative approach to efficient ensemble and lifelong learning." arXiv preprint arXiv:2002.06715 (2020).

**Summary Of The Paper:**

This paper tackles an important problem of the adversarial robustness of neural networks. Specifically, the authors of the paper propose a novel "kernel average pooling" layer that can be readily applied to most of the existing network architectures. Inspired by the recent success of neural network ensembles, the kernel average pooling layer applies a mean filter along the kernel dimension of the layer activation tensor. The authors of the paper also insights into why such a pooling layer can combat the problem of adversarial robustness, and conduct a series of experiments to demonstrate the effectiveness of the proposed method.

**Summary Of The Review:**

All in all, despite the interesting approach and observation of the proposed method, there are several limitations regarding the proposed approach. As such, I think the paper is marginally above the acceptance threshold.

---

> ### Author Response · Authors · 2022-11-15
> **KAP has lower clean accuracy but is highly scalable**
>
> We thank the reviewer for their feedback on our work. The reviewer raises concern about a) the lower clean accuracy of our proposed method compared to other baselines and b) inconsistency in robustness improvements. We argue that 1) while our proposed approach yields lower clean accuracy, this downside is largely compensated by its high scalability due to its substantially lower computational cost; 2) the baseline with noisy activations that the reviewer is referring to shows robustness against only a specific type of attack (SQUARE) and consistently achieves much lower accuracies against ensemble attacks such as AutoAttack. We believe our responses should address the reviewer’s concerns and welcome any additional discussions.
>
> - *"Overall, I find this to be an interesting approach. The main complaint I have regarding this approach is that the robustness comes at the cost of significantly lower clean data accuracy, and at the same time, the improvement of adversarial robustness is not consistent, as seen from the fact that regular ResNet18 model with activation noise can sometimes outperform the proposed method in terms of adversarial robustness."*
>
> *Lower clean accuracy*: We understand that lower clean accuracy is an undesired byproduct of our proposed approach and we have discussed this fact both in the results section as well as in the conclusion. However, we ask the reviewer to also consider the high scalability potential of our approach too. In particular: 1) as we report in Table A4, the training cost of our approach is only slightly higher than regular training of any neural network architecture; 2) our experiments show that increasing the depth and width of the network could further amplify the network robustness and thus scaling up the KAP network capacity could lead to further gains in robustness against adversarial attacks at a fraction of the computational cost associated with alternative approaches to robustness such as adversarial training. Together, we argue that these points already make a strong case for the attractiveness of our approach in practice.
>
> *Inconsistency in robustness*: Regarding the point on inconsistency of robustness, we want to highlight that the network with only activation noise could only improve the network robustness against the SQUARE attack and not significantly against PGD or AutoAttack while KAP networks with activation noise are consistently robust against all of these attacks. It is widely accepted that robustness against specific attacks should not be taken as proof of robustness and proper evaluations should consist of various approaches including white-box and black-box attacks as well as adaptive attacks that are tailored to specific approaches. In our evaluations, we closely followed these guidelines and evaluated all models against standard benchmark attacks (PGD and AutoAttack) as well as attacks tailored for approaches that include stochasticity (AutoAttack-random and SQUARE).
>
> - *"While there are some insights offered on why the proposed method can help adversarial robustness in section 3.4, most of the explanation is quite descriptive and lacks concrete details in my opinion. It might be good to go in-depth on this and offer some additional insights mathematically."*
>
> We believe that our approach can be understood as multiple models (KAP blocks) chained together, that each follow the theory of Lecuyer et al. 2019, Cohen et al. 2019, and Horvath et al. 2022 that we had cited in section 3.4. An advantage over the single noise injection approach of prior work is that we ensure sensitivity to noise is decreased throughout the entire network (since we reinject noise in every KAP block).
>
> - *“Maybe it makes sense to also benchmark the proposed method against some other cheap ensembling methods, such as [1][BatchEnsemble]?”*
>
> We thank the reviewer for suggesting this additional baseline. To address this comment, we trained a BatchEnsemble method using a public implementation of this method in PyTorch (https://github.com/giannifranchi/LP_BNN). We trained a WRN-16-4 BatchEnsemble model with an ensemble size 9 on the CIFAR10 dataset. This model was the largest WRN BatchEnsemble model with a comparable number of layers that could be fit on one GPU with 16GB memory. We chose an ensemble size of 9 to make it roughly comparable with our KAP model with kernel size of 3x3. It is worth mentioning that this model was substantially slower than the RN18 architecture that we had used in our analyses. This model achieved a clean accuracy of 95.4% and an AutoAttack ($\epsilon$=0.03) accuracy of 7.80% (see the updated Table-1 for full results). While like KAP, BatchEnsemble also forms ensembles of kernels at each layer, each ensemble is formed independent of other ensembles (i.e. no overlap between ensembles as implemented via  separate ensemble-specific slow weights and M=9 fast weights).
>
> [continued in next comment]

---

> > ### Author Response · Authors · 2022-11-15
> > **KAP has lower clean accuracy but is highly scalable (Cont'd)**
> >
> > [Continued from previous comment]
> >
> > - *"Other than adversarial robustness, does the proposed method offer other benefits? Does it help combat robustness against natural noise, model calibration, etc., since these are common benefits observed from a regular neural network ensemble? It might be interesting to investigate further other potential benefits as well."*
> >
> > To address this comment, we tested the performance of each of the models against natural perturbations from CIFAR-10-C dataset. We found that the RN18-NT model trained with Gaussian noise was consistently the best performing model on all of these benchmarks compared to both AT and RN18-KAP models which is potentially because of the significantly higher clean accuracy of this model. The KAP-RN18 model was better or on par with AT model on images perturbed with various noise distributions like Gaussian noise, Impulse noise, and Shot noise but was worse than AT model on more global perturbations such as motion blur and contrast.

---

### Official Review · Reviewer_nJ8u · 2022-10-26

**Confidence:** 4
**Correctness:** 2
**Technical Novelty And Significance:** 2
**Empirical Novelty And Significance:** 2
**Recommendation:** 3

**Clarity, Quality, Novelty And Reproducibility:**

- The KAP operation seems not novel. It is unclear how it contributes to improve model robustness.
- The technicality of KAP is not clear and it lacks strong discussions about KAP in improving robustness.
- It lacks the comparison to other baselines.

**Details Of Ethics Concerns:**

There is no ethics concerns.

**Strength And Weaknesses:**

Strength
- The proposed operation is simple and economic.

Weaknesses
- The paper is not well-presented and well-written.
   -  In the introduction section, it jumps direction into "our central premise in this work" to talk about ensemble without any hints or smooth transition.
   - Especially, the section about kernel average pool is not clear and solid. Equation (3) does not make sense to me because $w_i.x$ for all $i$. It is hard to interpret this operation. The authors should give more context about the shape of $x$, $z$ and so on. In Algorithm 1, it seems that KAP is average pooling over the depth. So it is not novel to me. Additionally, after applying average pooling over the depth, the output tensor gets smaller, how you can get back the shape W,H,D as in the last line of Algorithm 1.
- There are some vague arguments to me, for example "individual kernels within a network are robust". How can we justify if a kernel is robust?
- How to interpret and understand Equation (9) because I cannot see any random factor in KAP.
- The experiments are humble without comparing to other SOTA baselines.

**Summary Of The Paper:**

This paper proposes kernel average pooling operation to improve model robustness.

**Summary Of The Review:**

This paper considers KAP operation to improve model robustness. It says that KAP focuses on learning feature ensembles that form local “committees” similar to those used in Boosting and Random Forests. However, I cannot see how KAP realizes feature ensembles. It seems to me that KAP is average pooling over the depth.

---

> ### Author Response · Authors · 2022-11-15
> **Kernel Average Pooling is novel both as a neural net building block and in its application to adversarial robustness**
>
> We thank the reviewer for providing feedback on our submission. The reviewer raises several issues with 1) the clarity of our methodology; 2) the novelty of our approach; 3) absence of baselines. In our detailed responses below we argue that 1) majority of the comments about the clarity of our methodology were already answered in our original submission; 2) the reviewer’s opinion on lack of novelty of our approach needs to be substantiated by providing evidence of similar work and to the best of our knowledge, no similar approaches have previously been published in the adversarial ML literature; 3) Given the clear statement of our claim which compares our proposed approach with “adversarial training” method (and not claiming a new SOTA), comparing with SOTA baselines is unnecessary. Below we provide detailed responses for each individual point raised by the reviewer. We believe that these responses clarify all of the reviewers' concerns and we welcome any further points the reviewer may want to discuss.
>
> **1) Novelty**
>
> - *“The KAP operation seems not novel. It is unclear how it contributes to improve model robustness.”/ “ In Algorithm 1, it seems that KAP is average pooling over the depth. So it is not novel to me”*
>
> Sadly, the reviewer is clearly judging the novelty of our contribution by its simplicity. We strongly disagree with the reviewers’ perspective stating: “it seems that KAP is average pooling over the depth. So it is not novel to me.”. We respectfully insist that our proposed Kernel Average Pooling and its application to improving neural networks robustness is novel because of the following reasons: a) To the best of our knowledge, the proposed Kernel Average Pooling as explained in our manuscript has not been reported previously in the literature. b) Emergence of topographically organized kernels by using KAP in neural network layers has not been demonstrated before; c) The effect of KAP on network robustness has not been demonstrated before.
>
> **2) Clarity**
>
> - *“In the introduction section, it jumps direction into "our central premise in this work" to talk about ensemble without any hints or smooth transition.”*
>
> We thank the reviewer for bringing this lack of fluidity in our introduction. We will edit this section of the introduction to improve its readability.
>
> - *“In particular, the section about kernel average pool is not clear and solid. Equation (3) does not make sense to me because wi.x for all i. It is hard to interpret this operation. The authors should give more context about the shape of x, z and so on. In Algorithm 1, it seems that KAP is average pooling over the depth. So it is not novel to me. Additionally, after applying average pooling over the depth, the output tensor gets smaller, how you can get back the shape W,H,D as in the last line of Algorithm 1.”*
>
> In our submission, KAP operation, which is the core of our proposed approach, is described in three forms. First, it is described in section 3.2 with text and equations 2 and 3. Second, it is visualized in Figures 1 and A1. Third, it is algorithmically described in Algorithm 1.
>
> *Undefined variables*: The variable $x$ is defined in the first paragraph in section 3.1 and variable z is defined in the paragraph above equation 2.
>
> *Novelty*: See our comment regarding novelty below.
>
> *Output shape inconsistency*: As explained in Alg. 1, KAP implementation includes a padding operation which has the same functionality as padding in convolutional layers. We can control the output as desired using the padding function. For example, a padding of 1 for KAP kernel size of 3 and stride 1 produces an output tensor size with the same size as its input. However, similar to the way spatial pooling and convolutional layers function, the sizes do not have to match as long as the expected size for the next layer is set appropriately. In our experiments, we used “SAME” padding to ensure the same number of output kernels as the inputs are retained after every KAP layer.
>
> - *"There are some vague arguments to me, for example "individual kernels within a network are robust". How can we justify if a kernel is robust?"*
>
> Robustness of individual kernels can be measured the same way the robustness of a full network is assessed and this statement is the natural extension of applying the notion of robustness to perturbations as defined in eq. 1 to single features. Namely, robustness can be measured by quantifying how much each kernel’s output can be perturbed by applying small perturbations to the input. Intuitively, having individual kernels within a network to be robust means that the activation in the output of each kernel could not be drastically changed by making slight changes in its inputs.
>
> [continued in next comment]

---

> > ### Author Response · Authors · 2022-11-15
> > **Kernel Average Pooling is novel both as a neural net building block and in its application to adversarial robustness (Cont'd)**
> >
> > [Continued from previous comment]
> >
> > - *"How to interpret and understand Equation (9) because I cannot see any random factor in KAP."*
> >
> > We suspect that the reviewer may have misunderstood our approach in a fundamental way. As stated in the abstract, introduction, and section 3.4, the KAP operation and additive noise are independent components of our approach. Our empirical results also demonstrate the degree of robustness to adversarial attacks in KAP models with and without activation noise. As to equation 9, the term “$n_i$” in equation 9 corresponds to the random additive noise in the input of each block. The term $n_i$ is defined in the paragraph below equation 9. We hope that this explanation better clarifies the points made in our paper.
> >
> > - *"The technicality of KAP is not clear and it lacks strong discussions about KAP in improving robustness."*
> >
> > Please see above regarding our comments on lack of “technical clarity”. As for discussions about the role of KAP in improving robustness, in section 3.4, we have extensively discussed how including KAP in neural networks and its combination with activation noise leads to boosting adversarial robustness. Briefly, we draw links between KAP and activation noise to the literature on randomized smoothing with ensembles. Essentially, as we show in section 3.3, including KAP in layers of neural networks leads to emergence of kernel ensembles at each layer of the network and when these ensembles are combined with activation noise, they implement a recursive form of randomized smoothing across the layers of the network. We would appreciate any feedback from the reviewer on how to further improve these discussions.
> >
> > **3) Absence of  baselines**
> >
> > - *"The experiments are humble without comparing to other SOTA baselines."*
> >
> > Our statement of claims in the abstract, introduction, and throughout the paper clearly compares our proposed approach only with the commonly used adversarial training method. At no point in our manuscript we have claimed to have achieved a new state of art in terms of robustness. Adversarial training is a well-established method for improving robustness that has been used as reference in numerous prior work and as such constitutes a suitable point of reference as it has also been used in our work. In addition, recent work (Rice et al. 2020 and Gowal et al. 2020) have shown that adversarial training with early stopping is indeed one of the most reliable methods of robustness.
> > The significance of our approach is in the substantial boost in robustness that comes at a overwhelmingly lower computational cost compared to most alternative approaches to robustness. For these reasons, we believe that comparison to other SOTA baselines is not necessary.
> >
> > Rice, Leslie, Eric Wong, and Zico Kolter. "Overfitting in adversarially robust deep learning." International Conference on Machine Learning. PMLR, 2020.
> >
> > Gowal, S., Qin, C., Uesato, J., Mann, T., & Kohli, P. (2020). Uncovering the limits of adversarial training against norm-bounded adversarial examples. arXiv preprint arXiv:2010.03593.
> >
> >
> > - *"It lacks the comparison to other baselines."*
> >
> > We have included several baselines in our original submission including model with no KAP and no activation noise, model with no KAP and activation noise, model with KAP and no activation noise, and model with KAP and activation noise (Please see Tables 1 and 2 and Figure 2a). We expect that these baselines will be sufficient for making the deduction about the contributions of each of the components of our approach on improving robustness.

---

> ### Comment · Reviewer_nJ8u · 2022-12-05
> **After reading rebuttal**
>
> Thanks for your response. However, I am keen on my current scores. The writing needs a significant improvement. For instance, in Eq. (3), running variable $l$ has no role in the expression $w_i.x$. I cannot see how KAP realizes feature ensembles. To be more rigorous, KAP is average pooling over the depth. If we can consider it as feature ensembles, a standard Conv2D layer can also be considered as feature ensembles. Finally, according to Table 2, the proposed KAP significantly hurts the nat accuracy.

---

> > ### Author Response · Authors · 2022-12-07
> > **Kernel ensembles with similar functions only emerge in KAP networks**
> >
> > We thank the reviewer for their response to our rebuttal. Please see our further responses to your comments below.
> >
> >
> > - *”in Eq. (3), running variable l has no role in the expression wi.x.“*
> >
> > We thank the reviewer for bringing this issue to our attention. We will update this equation in our next revision. In its revised form, $w_i$ is replaced by $w_{il}$ and $x$ is replaced by $x_i$.
> >
> > - *”I cannot see how KAP realizes feature ensembles. To be more rigorous, KAP is average pooling over the depth. If we can consider it as feature ensembles, a standard Conv2D layer can also be considered as feature ensembles.”*
> >
> > Respectfully, the reviewer’s induction is incorrect in saying that if KAP produces feature [kernel] ensembles then a standard Conv2D layer would also produce feature [kernel] ensembles. We invite the reviewer to revisit section 3.3 in our manuscript where we have provided intuitive and mathematical support for how learning with gradient descent in a network with kernel average pooling leads to learning kernel ensembles with similar functions. Note that the provided support is only true for application of KAP and not generally true for any Conv2D operation with arbitrary weight parameters.
> >
> > - *”Finally, according to Table 2, the proposed KAP significantly hurts the nat accuracy.”*
> >
> > This comment is similar to reviewer **ckSs**’s comment in their first review to which we have already replied in our response to that reviewer. For completeness, we provide the same response here:
> >
> > *Lower clean accuracy*: We understand that lower clean accuracy is an undesired byproduct of our proposed approach and we have discussed this fact both in the results section as well as in the conclusion. However, we ask the reviewer to also consider the high scalability potential of our approach too. In particular: 1) as we report in Table A4, the training cost of our approach is only slightly higher than regular training of any neural network architecture; 2) our experiments show that increasing the depth and width of the network could further amplify the network robustness and thus scaling up the KAP network capacity could lead to further gains in robustness against adversarial attacks at a fraction of the computational cost associated with alternative approaches to robustness such as adversarial training. Together, we argue that these points already make a strong case for the attractiveness of our approach in practice.

---

### Decision · Program_Chairs · 2023-01-20

**Decision:**

Reject

**Justification For Why Not Higher Score:**

I listed 2 reasons above in my review:

+ Inconsistent empirical results, with even the regular model outperforming in adversarial robustness.
+ There's not much description behind why the strategy might work, either theoretically or with empirical intuition (e.g., toy examples). The paper makes several claims in general that aren't really backed up.

**Justification For Why Not Lower Score:**

N/A

**Metareview: Summary, Strengths And Weaknesses:**

This paper looks at improving adversarial robustness, motivated by ensemble-like strategies. In particular, the authors propose kernel average pooling, a layer for neural networks takes the average filter from the kernel dimension of intermediate features. The authors claim this produces functionality similar to an ensemble of kernels. They then validate their approach empirically against adversarial attacks on several image datasets.

There are two major complaints cited by reviewers that I particularly agree with.

+ Inconsistent empirical results, with even the regular model outperforming in adversarial robustness.
+ There's not much description behind why the strategy might work, either theoretically or with empirical intuition (e.g., toy examples). The paper makes several claims in general that aren't really backed up.

All reviewers leaned toward reject, and I agree with their opinion.